# A Wind Turbine Fault Classification Model Using Broad Learning System Optimized by Improved Pelican Optimization Algorithm

**Wumaier Tuerxun** [1,2], **Chang Xu** [3,*], **Muhaxi Haderbieke** [2], **Lei Guo** [1,4] **and Zhiming Cheng** [3]

1    College of Water Conservancy and HydroPower Engineering, HoHai University, Nanjing 210098, China; wapadar214@hhu.edu.cn (W.T.); guolei1982@hhu.edu.cn (L.G.)
2    College of Hydraulic and Civil Engineering, Xinjiang Agricultural University, Urumqi 830052, China; m8777360@126.com
3    College of Energy and Electrical Engineering, HoHai University, Nanjing 210098, China; 1705050223@hhu.edu.cn
4    School of Mechanical Engineering, Nanchang Institute of Technology, Nanchang 330099, China
*    Correspondence: zhuifengxu@hhu.edu.cn; Fax: +025-8378-6675

**Abstract:** As a classification model, a broad learning system is widely used in wind turbine fault diagnosis. However, the setting of hyperparameters for the models directly affects the classification accuracy of the models and it generally relies on practical experience and prior knowledge. In order to effectively solve the problem, the parameters of the broad learning system such as the number of feature nodes, the number of enhancement nodes, and the number of mapped features layer were optimized by the improved pelican optimization algorithm, and a classification model was built based on the broad learning system optimized by the improved pelican optimization algorithm. The classification accuracy of the proposed model was the highest and reached 98.75%. It is further shown that compared with the support vector machine, deep belief networks, and broad learning system models optimized by particle swarm optimization algorithm, the proposed model effectively improves the accuracy of wind turbine fault diagnosing.

**Keywords:** fault classification; wind turbine; broad learning system (BLS); pelican optimization algorithm (POA); parameter optimization





## 1. Introduction

As a clean renewable energy source, wind energy does not produce pollutants [1], and WT is an important contributor toward energy production free of $CO_2$ [2]. According to the most recent report of the WWEA, the global wind power capacity reached 840 Gigawatt in 2021 [3]. Wind farms are generally located in remote areas such as plateaus or coastal areas, with inconvenient transportation, a poor working environment, and susceptibility to extreme weather conditions, which bring many difficulties to the operation and maintenance of wind turbines [4]. With the increase in operation time, various types of failures may occur in the electrical system, transmission system, and control system of WT, leading to the WT's abnormal operation, and high operation and maintenance costs [5].

Fault diagnosis technology plays an important role in the running safety of WT, and maintenance staff use this technology to detect abnormality in time, handle it accordingly, and to extend the lifetime of WT [6]. An analysis of the series data obtained by a SCADA system is usually used to diagnose a WT's condition [7]. A large number of variables related to WT operating characteristics, such as wind speed, output power, temperature, current, and voltage are collected in the SCADA system, which can provide a rich source of data for WT diagnosis [8].

Artificial intelligence (AI) methods such as machine learning (ML), artificial neural network (ANN), and deep learning (DL) are commonly used in the WT fault diagnosis,

because of their robustness and self-adaptive capabilities. ML is to improve computer intelligence by independently learning the patterns that exist in large amounts of data and gaining new experience and knowledge to have a human-like decision-making ability. The ML algorithms that are most widely used in fault diagnosis are mainly K-nearest neighbor (KNN) [9], SVM [10], random forest (RF) [11], and extreme learning machine (ELM) [12]. ANN is a complex network that simulates the structure of the biological nervous system, with a large number of simple interconnected neurons. It performs the parallel processing of information and non-linear transformation by imitating the way a human brain neural processes information [13]. In the past decades, it has been developed rapidly and widely used in fault diagnosis of WT, for example, ANN methods have been used to diagnose faults in gearboxes [14–17] bearings [18,19], and generators [20,21].

Although ANN and ML have been broadly applied for fault diagnosis in recent decades, their diagnostic performance largely relies on feature extraction and selection. Instead, the DL algorithm has gradually become the main method for WT fault diagnosis due to its powerful feature extraction ability. At present, DL methods such as convolutional neural networks (CNN) [22,23], stacked automatic encoders (SAE), recurrent neural networks (RNN) [24], and deep belief networks (DBN) [25] are used in the fault diagnosis of rotating machinery.

Due to the complex deep structure of DL, which involves a large number of hyperparameters, the DL network is trained slowly and easily falls into a locally optimal solution. To overcome these problems, Chen et al. [26] proposed the BLS, which not only has a simple structure, fast training speed, and high accuracy rate but also has the advantage of enhanced learning. After BLS was proposed, researchers have successively proposed many improvements of BLS [27,28] and applied them to image classification [29], pattern recognition [30], numerical regression [31], automatic control [32], and other fields.

As a novel learning method, BLS solves the problem that the traditional neural network training methods are not applicable to multi-layer network training, effectively prevents the network from falling into local optimum, and improves the effectiveness of network convergence. In order to build a BLS model that satisfies the requirements, it is necessary to adapt the relevant parameters, and researchers often set the parameters according to their practical experience and a priori knowledge. For different problems, it may be necessary to repeatedly manually tune the relevant parameters.

To solve such problems, the parameters of BLS are optimized using the IPOA algorithm, hence, constructing the IPOA-BLS classification model. The experimental results indicate that the proposed model can better classify the faults of WT. The main contributions are highlighted as follows:

(1) The experimental SCAD data were preprocessed and fault-related features were selected based on the principal component analysis (PCA) method [33,34];
(2) The original POA algorithm was improved and was tested with benchmark functions;
(3) The parameters of the BLS such as the number of feature nodes $N_f$, number of enhancement nodes $N_e$, and number of mapped feature layers $N_{fl}$ were optimized by the IPOA algorithm;
(4) The processed WT fault data were used as the experimental sample with the SVM, DBN, BLS, PSO-BLS, POA-BLS, and IPOA-BLS models to classify and compare performance.

The rest of this paper is organized as follows: BLS, POA, and IPOA algorithms are introduced in Section 2. Section 3 describes SCADA data preprocessing, the IPOA-BLS classification model, and relevant parameter settings. Some graphical results with an analysis are given in Section 4. Finally, Section 5 concludes this paper.

## 2. Description of BLS, POA, and IPOA

### 2.1. BLS

BLS is a novel learning method proposed on the basis of a random vector functional-link neural network (RVFLNN) [35,36]. It does not require a time-consuming training

process and has a strong function approximation capability. The structure diagram of RVFLNN and BLS is shown in Figure 1.

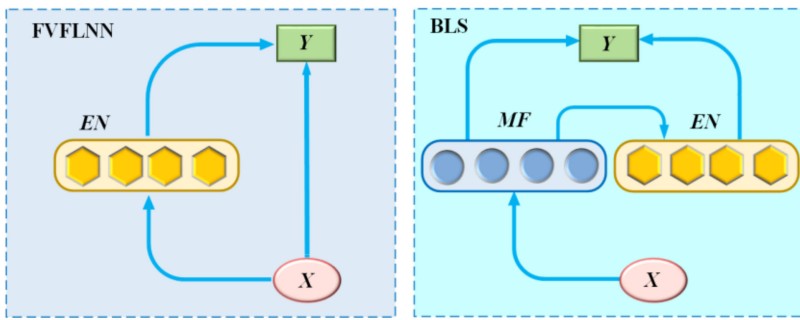

**Figure 1.** Structure of RVFLNN and BLS.

As can be seen from Figure 1, RVFLNN consists of three parts, namely the input layer, the enhancement node, and the output layer, and the output layer connects both the input layer and the enhancement nodes (EN). In contrast to RVFLNN, the BLS maps to the mapped feature (MF) first before the input layer data is mapped to EN, and the output layer connects both MF and EN.

In terms of network structure, the network of BLS is horizontally expanded and vertically fixed, which is very different from deep neural networks, and the number of layers is greatly reduced, and its structure is shown in Figure 2. A feature node $Z_i$, maps the original data to obtain the following formula:

$$Z_i = \theta_i\left(XW_{fi} + B_{fi}\right), \ \ i = 1, 2, \ldots, n \tag{1}$$

where $X$ is the input layer data; $W_{fi}$ and $B_{fi}$ denote the weight and bias of the input layer $X$ to $Z_i$, respectively, which is generally fine-tuned by a sparse autoencoder to produce the optimal value; $\theta_i$ is a linear or nonlinear activation function; $Z_i$ is the $i$-th group of feature nodes containing $p$ neurons.

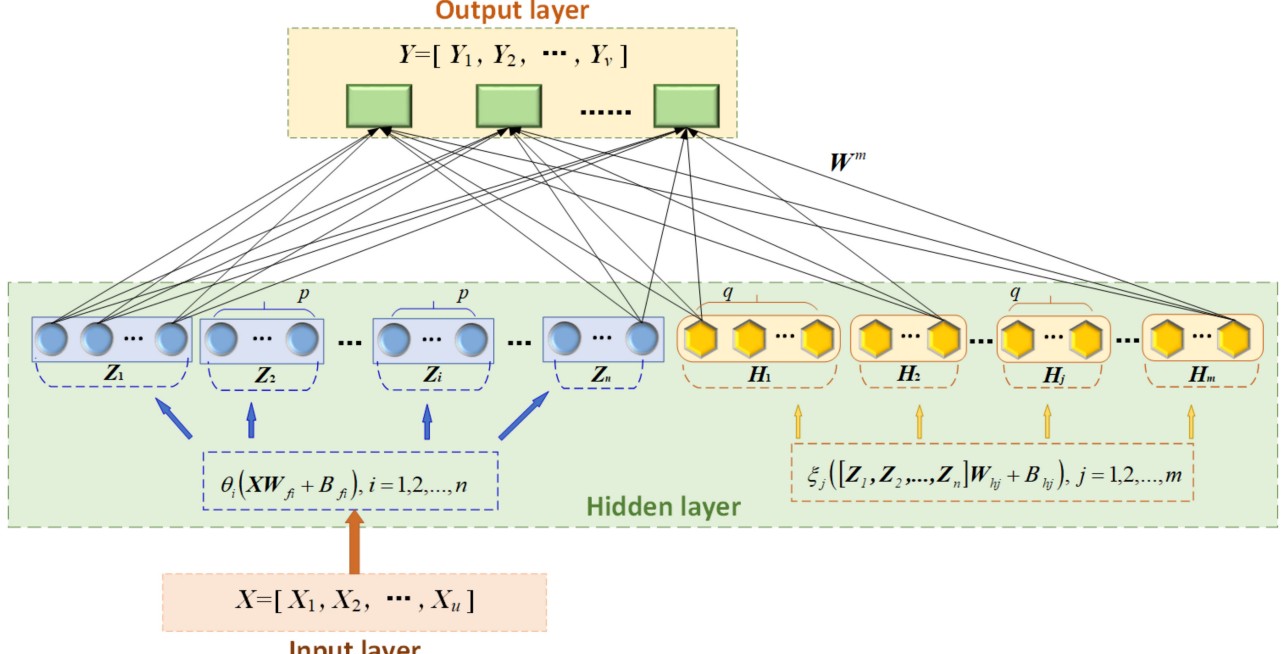

**Figure 2.** Structure of BLS.

All feature nodes can be represented as $\mathbf{Z}^n = [\mathbf{Z}_1, \mathbf{Z}_2, \ldots, \mathbf{Z}_i, \ldots, \mathbf{Z}_n]$, after obtaining the feature node $\mathbf{Z}^n$, it can be used to calculate the enhancement node $\mathbf{H}^m$.

$$
\begin{aligned}
\mathbf{H}_j &= \xi_j\left(\mathbf{Z}^n \mathbf{W}_{hj} + \mathbf{B}_{hj}\right), \quad j = 1, 2, \ldots, m \\
\mathbf{H}^m &= \left[\mathbf{H}_1, \mathbf{H}_2, \ldots, \mathbf{H}_j, \ldots, \mathbf{H}_m\right]
\end{aligned}
\tag{2}
$$

where $\mathbf{H}_j$ is the $j$-th group of enhancement nodes containing $q$ neurons; $\xi_j$ is the nonlinear activation function; $\mathbf{W}_{hj}$ and $\mathbf{B}_{hj}$ represent weights and biases, respectively. The model of the BLS can be obtained by further calculations:

$$
\begin{aligned}
\mathbf{Y} &= [\mathbf{Z}^n | \xi(\mathbf{Z}^n \mathbf{W}_{h1} + \mathbf{B}_{h1}), \ldots, \xi(\mathbf{Z}^n \mathbf{W}_{hm} + \mathbf{B}_{hm})] \mathbf{W}^m \\
&= [\mathbf{Z}^n | \mathbf{H}_1, \ldots, \mathbf{H}_m] \mathbf{W}^m \\
&= [\mathbf{Z}^n | \mathbf{H}^m] \mathbf{W}^m \overset{\text{if } A = [\mathbf{Z}^n | \mathbf{H}^m]}{\longrightarrow} \mathbf{Y} = \mathbf{A} \mathbf{W}^m
\end{aligned}
\tag{3}
$$

where $\mathbf{W}^m$ represents the weight between the output layer and the hidden layer formed by the feature nodes and enhancement nodes. In the BLS, parameters such as $\mathbf{W}_{fi}$, $\mathbf{B}_{fi}$, $\mathbf{W}_{hj}$, $\mathbf{B}_{hj}$ are randomly generated and fine-tuned by a sparse autoencoder, and remain constant during the training process; the metrics learned in the network are only $\mathbf{W}^m$. The objective function of the BLS is:

$$
\arg \min_{\mathbf{W}^m} : \|\mathbf{A}\mathbf{W}^m - \mathbf{Y}\|_2^2 + \lambda \|\mathbf{W}^m\|_2^2 \Rightarrow \mathbf{W}^m = \left(\mathbf{A}^T \mathbf{A} + \lambda \mathbf{I}\right)^{-1} \mathbf{A}^T \mathbf{Y}
\tag{4}
$$

where $\|\mathbf{A}\mathbf{W}^m - \mathbf{Y}\|_2^2$ is used to control the minimization of training error; $\lambda \|\mathbf{W}^m\|_2^2$ is used to prevent overfitting of the model; $\lambda$ is the regularization factor; $\mathbf{Y}$ is the output; $\mathbf{A}^T$ is the transpose matrix of $\mathbf{A}$; $\mathbf{I}$ is the unit matrix.

### 2.2. POA

The POA is a new meta-heuristic optimization algorithm [37] inspired by pelican hunting behaviors. It has the advantages of few adjustment parameters, fast convergence speed, and simple calculation. Pelicans are found in the warm waters of the world and live mainly in lakes, rivers, coasts, and swamps [38]. Pelicans generally live in flocks; they are not only good at flying but also good at swimming [39]. They have sharp eyesight in flight, as well as excellent observation skills, and they mainly feed on fish. After determining the location of the prey, the pelicans rush towards the prey from a height of 10–20 m and dive straight into the water to hunt [40]. If pelicans find schools of fish, they will arrange themselves in a line or U-shape to swoop down from the sky towards the fish in the water and use their wings to flap the water, forcing fishes to move upwards, and then collect the fishes in their throat pouch. Depending on the above explanation, the mathematical model of the POA algorithm was built.

(1) Initialization: Assuming that there are $N$ pelicans in an $M$ dimensional space, the position of the $i$-th pelican in the $M$ dimensional space is $P_i = [p_{i1}, p_{i2}, \ldots, p_{im}, \ldots, p_{iM}]$, the position $\mathbf{P}$ of the $N$ pelicans is expressed as follows:

$$
\mathbf{P} = \begin{bmatrix} \mathbf{P}_1 \\ \mathbf{P}_2 \\ \vdots \\ \mathbf{P}_i \\ \vdots \\ \mathbf{P}_N \end{bmatrix} = \begin{bmatrix} p_{11} & p_{12} & \cdots & p_{1m} & \cdots & p_{1M} \\ p_{21} & p_{22} & \cdots & p_{2m} & \cdots & p_{2M} \\ \vdots & \vdots & \vdots & \vdots & & \vdots \\ p_{i1} & p_{i2} & \cdots & p_{im} & \cdots & p_{iM} \\ \vdots & \vdots & \vdots & \vdots & & \vdots \\ p_{N1} & p_{N2} & \cdots & p_{Nm} & \cdots & p_{NM} \end{bmatrix}, \ i = 1, 2, \ldots\ldots, N
\tag{5}
$$

where $p_{im}$ denotes the position of the *i*-th pelican in the *m*-th dimension. At the initialization stage, pelicans are randomly distributed within a certain range, and the position update of the pelican is described as

$$P_{im} = low_m + random \cdot (up_m - low_m), \ i = 1, 2, \ldots, N; \ m = 1, 2, \ldots, M; \tag{6}$$

where $low_m, up_m$ is the search range of the pelican; *random* is a random number between (0, 1).

(2) Moving towards prey: In this phase, the pelican identifies the prey's location and then rushes to the prey from a high altitude as shown in Figure 3. The random distribution of the prey in the search space increases the exploration ability of the pelican, and the update of the pelican's location during each iteration is described as

$$\boldsymbol{P}_{im}^{t+1} = \begin{cases} \boldsymbol{P}_{im}^t + rand \cdot (S_m^t - \lambda \cdot \boldsymbol{P}_{im}^t), & F(P_S) < F(P_i) \\ \boldsymbol{P}_{im}^t + rand \cdot (\boldsymbol{P}_{im}^t - S_m^t), & F(P_S) \geq F(P_i) \end{cases} \tag{7}$$

where *t* is a current iteration number; $\boldsymbol{P}_{im}^t$ denotes the position of the *i*-th pelican in the *m*-th dimension; $S_m^t$ is the position of the prey in the *m*-th dimension; $\lambda$ is randomly equal to 1 or 2; $F(P_s)$ is the objective function value; $F(P_i)$ denotes the value of the fitness function of *i*-th pelican in the *m*-th dimension.

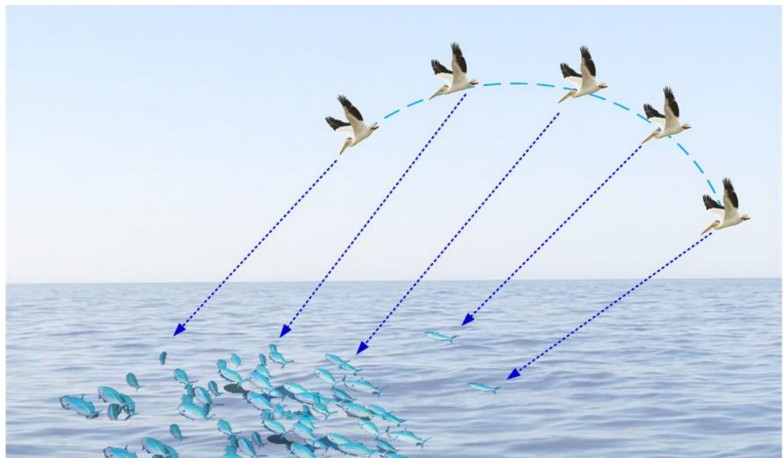

**Figure 3.** Diagram of the pelicans moving towards prey.

(3) Winging on the water surface: After the pelicans reach the surface of the water, they spread their wings on the surface of the water to move the fish upwards, then collect the prey in their throat pouch. This behavior of pelicans during hunting is mathematically simulated

$$\boldsymbol{P}_{im}^{t+1} = \boldsymbol{P}_{im}^t + \gamma \cdot \left( \frac{T - t}{T} \right) \cdot (2 \cdot random - 1) \cdot \boldsymbol{P}_{im}^t \tag{8}$$

where *t* is a current number of iterations; *T* is a maximum iteration number; $\gamma \cdot \left( \frac{T-t}{T} \right)$ is the neighborhood radius of $\boldsymbol{P}_{im}^t$, and it represents the radius of the neighborhood of the population members to search locally near each member to converge to a better solution; *random* is a random number between (0, 1).

### 2.3. IPOA

After improving the original POA algorithm, the optimization efficiency could be further improved. The specific improvement strategy is as follows.

(1) Initialization strategy: The Tent chaotic map [41] is used to replace the randomly generated method in the original POA to initialize the pelicans after the Tent chaotic mapping is introduced, and the Equation (6) can be rewritten as follows:

$$p_{im} = low_m + Tent \cdot (up_m - low_m), \ i = 1, 2, \ldots, N; \ m = 1, 2, \ldots, M; \tag{9}$$

$$Tent^{t+1} = \begin{cases} \frac{Tent^t}{z}, & Tent^t \in [0, z) \\ \frac{(1 - Tent^t)}{(1-z)}, & Tent^t \in [z, 1] \end{cases} \tag{10}$$

where $t$ is a current number of iterations; $T$ is a maximum iteration number; where $z \in (0, 1)$, $Tent^t \in [0, 1]$, $t = 1, 2, \ldots, T$.

At this stage, the position of the pelicans is initialized using the Tent chaotic map, which helps to improve the global search performance of the POA algorithm.

(2) Moving towards prey: At this stage, the dynamic weight factor [42] $\theta$ helps the pelican to constantly update its position. At the beginning of the iteration, $\theta$ has a large value, when the pelican is able to perform a better global search, and at the end of the iteration $\theta$ decreases adaptively, and this time the pelican is able to perform a better local search while increasing the convergence speed. the Equation (7) can be rewritten as follows:

$$P_{im}^{t+1} = \begin{cases} \theta = \frac{e^{2(1-t/T)} - e^{-2(1-t/T)}}{e^{2(1-t/T)} + e^{-2(1-t/T)}} \\ P_{im}^t + rand \cdot (S_m^t - P_{im}^t) \cdot \theta, & F(P_S) < F(P_i) \\ P_{im}^t + rand \cdot (P_{im}^t - S_m^t) \cdot \theta, & F(P_S) \geq F(P_i) \end{cases} \tag{11}$$

To prove the correctness and validity of the IPOA algorithm, ten benchmark functions were tested using PSO, GWO, WOA, POA, and IPOA. The value of each algorithm parameter is shown in Table 1.

**Table 1.** Parameter setting of PSO, GWO, WOA, POA, and IPOA.

| Algorithm | Parameter Setting |
|:---:|:---:|
| Common setting | Maximum iteration T = 1000 |
| | Population number: N = 30 |
| | Runs: r = 30 |
| PSO | C1 = 2 |
| | C2 = 2 |
| | W = 0.7 |
| GWO | m1 = 0.3 |
| WOA | L1 = 3 |
| | L2 = 5 |
| POA | $\Lambda = round \ (random \ (1,2))$ |
| | $\gamma = 0.2$ |
| IPOA | $\lambda = \theta$ |
| | $\gamma = 0.2$ |

Descriptions of the benchmark functions are shown in Table 2. In this table, $f\,1$–$f\,4$ are the unimodal test functions, $f\,5$–$f\,7$ are the multimodal test functions, and $f\,8$–$f\,10$ are the composite benchmark functions.

**Table 2.** Benchmark functions.

| Benchmark Function | Function Name | D | Range | $f_{opt}$ |
|---|---|---|---|---|
| (1) Unimodal test functions | | | | |
| $f_1(x) = \Sigma_{i=1}^{n} x_i^2$ | Sphere | 30 | $[-100,100]$ | 0 |
| $f_2(x) = \sum_{i=1}^{D} |x_i| + \Pi_{i-1}^{D} |x_i|$ | Schwefel 2.22 | 30 | $[-10,10]$ | 0 |
| $f_3(x) = \sum_{i=1}^{D} \left( \Sigma_{j-1}^{D} x_i \right)^2$ | Schwefel 1.2 | 30 | $[-100,100]$ | 0 |
| $f_4(x) = max_i \{ |X_i|, 1 \leq i \leq D \}$ | Schwefel 2.21 | 30 | $[-100,100]$ | 0 |
| (2) Multimodal test functions | | | | |
| $f_5(x) = \sum_{i=1}^{D} \left( x_i^2 - 10 \cos(2\pi x_i) + 10 \sin(2\pi x_i) \right)$ | Rastrigin | 30 | $[-5.12,5.12]$ | 0 |
| $f_6(x) = 20 + e - 20 \exp\left( -20\sqrt{(1/D)\Sigma_{i=1}^{D} x_i^2} \right) - \exp((1/D)\sum_{i=1}^{D} \cos(2\pi x_i))$ | Ackley | 30 | $[-32,32]$ | $8.8818 \times 10^{-16}$ |
| $f_7(x) = (1/4000)\sum_{i=1}^{D}(x_i^2) - \left( \prod_{i=1}^{D} \cos\left( x_i/\sqrt{i} \right) \right) + 1$ | Griewank | 30 | $[-600,600]$ | 0 |
| (3) composite benchmark functions | | | | |
| $f_8(x) = \left( (1/500) + \sum_{j=1}^{25} \left( 1/\left( j + \sum_{i=1}^{2} (x_i - a_{ij})^6 \right) \right) \right)^{-1}$ | Foxholes | 2 | $[-65.53,65.53]$ | 0.998004 |
| $f_9(x) = \sum_{i=1}^{11} \left( a_i - \left( x_1 \left( b_i^2 + b_i x_2 \right)/b_i^2 + b_i x_3 + x_4 \right) \right)^{-1}$ | Six hump | 4 | $[-5,5]$ | 0.0003075 |
| $f_{10}(x) = -\sum_{i=1}^{10} \left[ (X - a_i)(X - a_i)^T + c_i \right]^{-1}$ | Langerman 10 | 4 | $[0,10]$ | $-10.5364$ |

Tables 3–5 report the relevant statistical experiment results from running each algorithm 30 times in the benchmark function independently. In these tables, "Best", "Worst", "Average", and "Std." represent the best value, the worst value, the average value, and the standard deviation of the algorithm, respectively.

**Table 3.** Statistical results for unimodal benchmark functions.

| Function | Statistics | PSO | GWO | WOA | POA | IPOA |
|---|---|---|---|---|---|---|
| $f1$ | Best | $8.0606 \times 10^{+1}$ | $1.3824 \times 10^{-58}$ | $1.3904 \times 10^{-130}$ | $8.1414 \times 10^{-207}$ | 0 |
| | Worst | $2.1442 \times 10^{+4}$ | $6.7429 \times 10^{+4}$ | $7.2836 \times 10^{+4}$ | $7.1909 \times 10^{+4}$ | $1.1224 \times 10^{+4}$ |
| | Average | $1.4696 \times 10^{+3}$ | $3.3884 \times 10^{+2}$ | $4.0963 \times 10^{+2}$ | $1.4462 \times 10^{+2}$ | $1.0947 \times 10^{+2}$ |
| | Std. | $2.6760 \times 10^{+3}$ | $3.4928 \times 10^{+3}$ | $4.1509 \times 10^{+3}$ | $2.5477 \times 10^{+3}$ | $1.3498 \times 10^{+3}$ |
| $f2$ | Best | $2.7469 \times 10^{+1}$ | $8.8984 \times 10^{-35}$ | $3.7388 \times 10^{-95}$ | $2.3633 \times 10^{-107}$ | 0 |
| | Worst | $5.3174 \times 10^{+12}$ | $7.3390 \times 10^{+12}$ | $7.2151 \times 10^{+13}$ | $3.6333 \times 10^{+13}$ | $5.9060 \times 10^{+12}$ |
| | Average | $1.2499 \times 10^{+10}$ | $7.3536 \times 10^{+9}$ | $7.3708 \times 10^{+10}$ | $3.6522 \times 10^{+10}$ | $5.9060 \times 10^{+8}$ |
| | Std. | $2.0235 \times 10^{+11}$ | $2.3208 \times 10^{+11}$ | $2.2820 \times 10^{+12}$ | $1.1490 \times 10^{+12}$ | $1.8676 \times 10^{+11}$ |
| $f3$ | Best | $9.8883 \times 10^{+3}$ | $6.1321 \times 10^{-14}$ | $3.0153 \times 10^{+4}$ | $1.8928 \times 10^{-204}$ | 0 |
| | Worst | $6.4994 \times 10^{+4}$ | $1.4551 \times 10^{+5}$ | $1.6016 \times 10^{+5}$ | $1.6112 \times 10^{+5}$ | $1.9157 \times 10^{+4}$ |
| | Average | $1.5945 \times 10^{+4}$ | $1.6766 \times 10^{+3}$ | $6.6527 \times 10^{+4}$ | $5.8836 \times 10^{+2}$ | $2.1205 \times 10^{+2}$ |
| | Std. | $6.8243 \times 10^{+3}$ | $9.3954 \times 10^{+4}$ | $2.9123 \times 10^{+4}$ | $7.8845 \times 10^{+3}$ | $6.0947 \times 10^{+3}$ |
| $f4$ | Best | $4.5408$ | $1.5182 \times 10^{-14}$ | $3.8109 \times 10^{+1}$ | $2.1908 \times 10^{-105}$ | $2.6256 \times 10^{-308}$ |
| | Worst | $5.9368 \times 10^{+1}$ | $8.8048 \times 10^{+1}$ | $8.8640 \times 10^{+1}$ | $8.7096 \times 10^{+1}$ | $9.5583 \times 10^{+1}$ |
| | Average | $1.2826 \times 10^{+1}$ | $1.8448$ | $5.0235 \times 10^{+1}$ | $3.5377 \times 10^{-1}$ | $1.1376 \times 10^{-1}$ |
| | Std. | $9.5704$ | $9.5534$ | $1.4559 \times 10^{+1}$ | $3.9212$ | $3.0484$ |

**Table 4.** Statistical results for multimodal benchmark functions.

| Function | Statistics | PSO | GWO | WOA | POA | IPOA |
|---|---|---|---|---|---|---|
| $f\,5$ | Best | $1.9909 \times 10^{+2}$ | $2.3932 \times 10^{-1}$ | 0 | 0 | 0 |
| | Worst | $4.4057 \times 10^{+2}$ | $4.4237 \times 10^{+2}$ | $4.4746 \times 10^{+2}$ | $4.5905 \times 10^{+2}$ | $5.6918 \times 10^{+2}$ |
| | Average | $2.4516 \times 10^{+2}$ | $1.4587 \times 10^{+1}$ | $1.3139 \times 10^{+1}$ | 4.4417 | $8.4287 \times 10^{-1}$ |
| | Std. | $4.0864 \times 10^{+1}$ | $5.2324 \times 10^{+1}$ | $4.9471 \times 10^{+1}$ | $3.0706 \times 10^{+1}$ | $1.8966 \times 10^{+1}$ |
| $f\,6$ | Best | 8.0718 | $1.6046 \times 10^{-14}$ | $3.9672 \times 10^{-15}$ | $4.2040 \times 10^{-15}$ | $8.8818 \times 10^{-16}$ |
| | Worst | $2.0233 \times 10^{+1}$ | $2.0699 \times 10^{+1}$ | $2.0725 \times 10^{+1}$ | $2.0740 \times 10^{+1}$ | $2.1099 \times 10^{+1}$ |
| | Average | $1.1201 \times 10^{+1}$ | $3.9652 \times 10^{-1}$ | $4.3575 \times 10^{-1}$ | $1.6615 \times 10^{-1}$ | $3.6062 \times 10^{-2}$ |
| | Std. | 2.7873 | 2.2655 | 2.1693 | 1.4060 | $7.2128 \times 10^{-01}$ |
| $f\,7$ | Best | 1.7694 | $2.7510 \times 10^{-3}$ | $6.1431 \times 10^{-3}$ | 0 | 0 |
| | Worst | $1.4914 \times 10^{+2}$ | $6.2027 \times 10^{+2}$ | $6.3457 \times 10^{+2}$ | $6.5070 \times 10^{+2}$ | $1.0007 \times 10^{+3}$ |
| | Average | $1.2120 \times 10^{+1}$ | 3.1599 | 3.9032 | 1.4482 | 1.0140 |
| | Std. | $1.8168 \times 10^{+1}$ | $3.2284 \times 10^{+1}$ | $3.7483 \times 10^{+1}$ | $2.3843 \times 10^{+1}$ | $3.1645 \times 10^{+1}$ |

**Table 5.** Statistical results for composite benchmark functions.

| Function | Statistics | PSO | GWO | WOA | POA | IPOA |
|---|---|---|---|---|---|---|
| $f\,8$ | Best | $9.9800 \times 10^{-1}$ | 4.7160 | 3.6734 | 1.0311 | 1.5871 |
| | Worst | $4.4687 \times 10^{+1}$ | $1.8978 \times 10^{+2}$ | $2.3138 \times 10^{+2}$ | $1.7590 \times 10^{+2}$ | $1.4355 \times 10^{+2}$ |
| | Average | 1.2597 | 5.0814 | 4.3297 | 1.3977 | 1.0184 |
| | Std. | 1.6981 | 6.1179 | 7.7406 | 5.9295 | 4.6540 |
| $f\,9$ | Best | $9.8558 \times 10^{-3}$ | $5.7168 \times 10^{-3}$ | $6.9056 \times 10^{-4}$ | $1.7361 \times 10^{-3}$ | $4.3471 \times 10^{-4}$ |
| | Worst | $2.0059 \times 10^{-1}$ | $1.9738 \times 10^{-1}$ | $3.7275 \times 10^{-1}$ | $8.3830 \times 10^{-1}$ | $8.4090 \times 10^{-1}$ |
| | Average | $1.0510 \times 10^{-2}$ | $6.1706 \times 10^{-3}$ | $1.7122 \times 10^{-3}$ | $3.4990 \times 10^{-3}$ | $1.5492 \times 10^{-3}$ |
| | Std. | $7.2294 \times 10^{-3}$ | $6.3856 \times 10^{-3}$ | $1.3101 \times 10^{-2}$ | $3.0524 \times 10^{-2}$ | $2.6717 \times 10^{-2}$ |
| $f\,10$ | Best | $-1.0177 \times 10^{+1}$ | $-1.0357 \times 10^{+1}$ | $-7.9845$ | $-9.6350$ | $-1.0520 \times 10^{+1}$ |
| | Worst | $-7.7636 \times 10^{-1}$ | $-7.4272 \times 10^{-1}$ | $-7.2858 \times 10^{-1}$ | $-7.6370 \times 10^{-1}$ | $-6.1054 \times 10^{-1}$ |
| | Average | $-9.1388$ | $-7.8546$ | $-7.2752$ | $-9.3844$ | $-9.6476$ |
| | Std. | 1.6680 | 2.1716 | 1.1787 | $8.9320 \times 10^{-1}$ | $1.5835 \times 10^{-1}$ |

The convergence curve of the *f 1–f 10* functions is shown in Figure 4: the red line denotes the IPOA algorithm, and the blue line represents the POA algorithm. It implies that the fitness value of the IPOA algorithm is lower and the convergence speed faster than other algorithms.

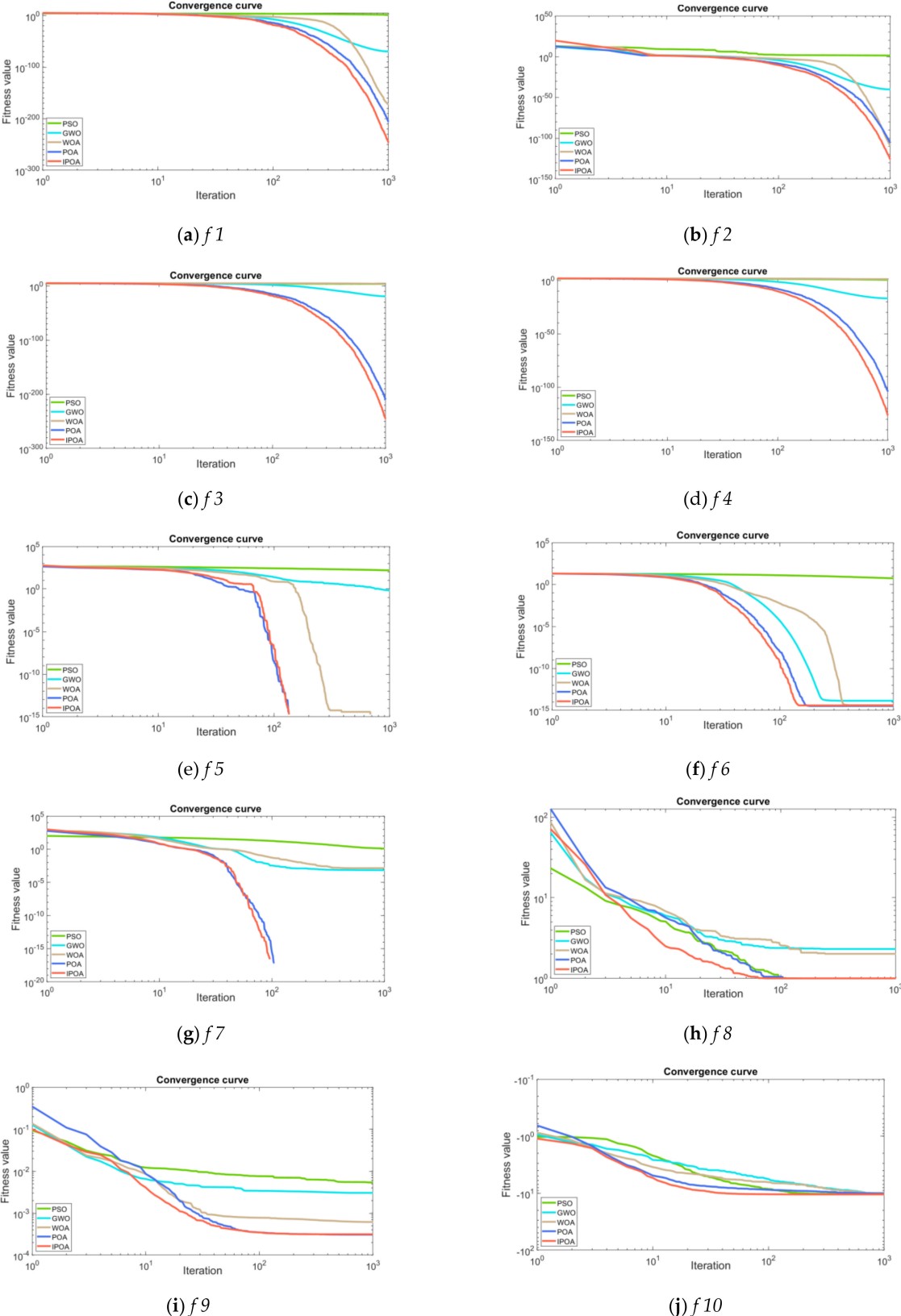

**Figure 4.** The convergence curves of benchmark functions.

## 3. Wind Turbine Fault Classification

### 3.1. SCADA Data Preprocessing

In this study, the information from the SCADA system recorded from 38 Vestas 1.5 MW wind turbines in a wind farm in Liu He from 1 January 2018 to 31 December 2018 was used as experimental data and the time resolution of the data was ten minutes. After data cleaning, the fault categories were coded in the experimental data set such as the gearbox overheated (F1), yaw system fault (F2), generator overheated (F3), inverter system fault (F4), and pitch system fault (F5). Each fault includes 640 training samples and 160 test samples. A total of 4000 samples were selected, among them 3200 samples for training and 800 samples for testing.

In this experimental SCADA data, there were 52 observation feature parameters of the Vestas wind turbine. After feature extraction with the PCA method and comparison with Pearson correlation coefficient values, 35 characteristic parameters were chosen as the observation parameters of fault diagnosis. The relevant feature codes and their description are specified in Table 6. The feature code and its Pearson correlation coefficient of each fault are shown in Table 7.

**Table 6.** Feature code and its description.

| Feature Code | Feature Description | Feature Code | Feature Description |
|---|---|---|---|
| X1 | impeller speed (rpm) | X19 | pitch bearings temperature (°C) |
| X2 | generator active power(kVA) | X20 | pitch oil pressure (mPa) |
| X3 | average reactive power (kVA) | X21 | pitch system motor speed (rpm) |
| X4 | average blade angle (deg) | X22 | nacelle temperature (°C) |
| X5 | bias of blade (m) | X23 | converter cabinet temperature (°C) |
| X6 | blade temperature (°C) | X24 | converter cabinet fan temperature (°C) |
| X7 | paddle motor temperature (°C) | X25 | yaw bearing radial force (kN) |
| X8 | paddle system temperature (°C) | X26 | yaw drive temperature (°C) |
| X9 | ambient temperature (°C) | X27 | yaw bearing axial force (kN) |
| X10 | generator stator winding temperature u (°C) | X28 | average wind speed (m/s) |
| X11 | generator stator winding temperature v (°C) | X29 | yaw inverter cabinet temperature (°C) |
| X12 | generator stator winding temperature w (°C) | X30 | yaw bearings temperature (°C) |
| X13 | stator temperature (°C) | X31 | yaw control cabinet temperature (°C) |
| X14 | rotor temperature (°C) | X32 | wind direction angle |
| X15 | rotor speed (rpm) | X33 | transformer temperature (°C) |
| X16 | gearbox oil temperature (°C) | X34 | generator torque (kN·m) |
| X17 | gearbox bearing temperature (°C) | X35 | tower temperature (°C) |
| X18 | gearbox oil pressure (mPa) | | |

**Table 7.** Feature code (Fc) and its Pearson correlation coefficient (Pcc) of each fault.

| Fc | Pcc | Fc | Pcc | Fc | Pcc | Fc | Pcc | Fc | Pcc |
|---|---|---|---|---|---|---|---|---|---|
| | | | | gearbox overheated (F1) | | | | | |
| X2 | 0.3872 | X9 | 0.1736 | X10 | 0.4436 | X11 | 0.4507 | X12 | 0.4523 |
| X13 | 0.3154 | X14 | 0.3511 | X15 | 0.4625 | X16 | 0.7831 | X17 | 0.8865 |
| X18 | 0.3221 | X22 | 0.1023 | X28 | 0.4178 | | | | |

**Table 7.** *Cont.*

| Fc | Pcc | Fc | Pcc | Fc | Pcc | Fc | Pcc | Fc | Pcc |
|---|---|---|---|---|---|---|---|---|---|
| | | | | yaw system fault (F2) | | | | | |
| X2 | 0.3661 | X25 | 0.3622 | X26 | 0.6134 | X27 | 0.3531 | X28 | 0.3164 |
| X29 | 0.5171 | X30 | 0.5731 | X31 | 0.5463 | X32 | 0.5273 | | |
| | | | | generator overheated (F3) | | | | | |
| X2 | 0.4175 | X3 | 0.4336 | X10 | 0.6734 | X11 | 0.6804 | X12 | 0.6779 |
| X13 | 0.7863 | X14 | 0.7162 | X15 | 0.5153 | X22 | 0.1227 | X34 | 0.5271 |
| | | | | converter system fault (F4) | | | | | |
| X14 | 0.4267 | X22 | 0.1103 | X23 | 0.6832 | X24 | 0.6054 | X33 | 0.3762 |
| X35 | 0.1272 | | | | | | | | |
| | | | | pitch system fault (F5) | | | | | |
| X1 | 0.3874 | X2 | 0.3642 | X25 | 0.4623 | X26 | 0.6723 | X27 | 0.4167 |
| X28 | 0.3465 | X29 | 0.5732 | X30 | 0.6264 | X31 | 0.6114 | X32 | 0.4178 |

*3.2. IPOA-BLS Classification Model*

As shown in Figure 5, the classification process of the proposed model is described as follows:

Step 1: The relevant data of WT were collected by the SCADA system.

Step 2: PCA was used to reduce the dimensionality of the preprocessed SCADA data, and fault-related features were selected. After cleaning and dimensionality reduction, 4000 groups were taken as experimental samples, and each sample had 35 features.

Step 3: Training samples and testing samples were taken from the experimental samples. In Figure 5, each of the five fault types included 640 training samples and 160 testing samples. $X_{i,j}$ represents the $j$-th feature value of the $i$-th sample, and $Y_{i,j}$ represents the $j$-th fault type of the $i$-th sample.

Step 4: The IPOA-BLS model was used for training and testing, respectively, and in this step:

(1) The BLS $\left\{N_f, N_e, N_{fl}\right\}$ parameter and the IPOA algorithm parameters, including the pelican's population $N$, the maximum iterations $T$, the upper limit of argument $up$, the lower limit of argument $low$, the dimension $M$, and the sample $data$ were initialized.

(2) The data of the fitness value were calculated. The classification accuracy obtained by the BLS was used as the fitness value, within the iteration range. Equations (9), (11), and (8) were used to calculate the location of the pelican. If the current new location was better, the old location was updated.

(3) According to the optimal parameter combination $\left\{N_f, N_e, N_{fl}\right\}$, the BLS was trained, and testing samples were used to classify and save the result.

Step 5: The results from Step 4 were anti-normalized.

Step 6: The final classification results were obtained and outputted with graphical.

Step 7: The relevant evaluation metrics were calculated.

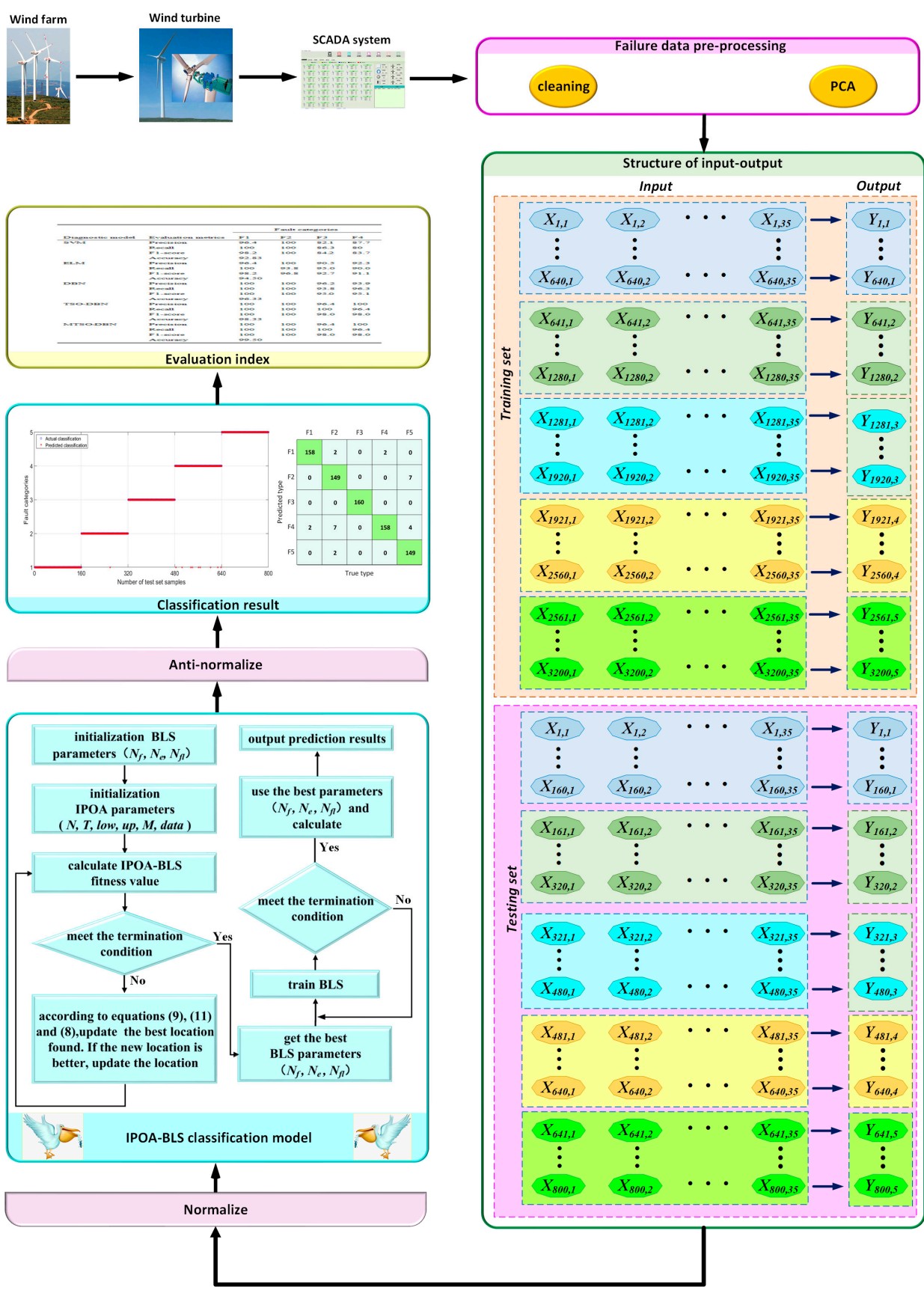

**Figure 5.** Classification process of the IPOA-BLS model.

### 3.3. Parameters Setting

In this paper, algorithms such as PSO, POA, and IPOA were used to optimize the hyperparameters of the BLS, and the relevant parameter values are shown in Table 8.

**Table 8.** The parameter values of PSO-BLS, POA-BLS, and IPOA-BLS.

| Model Name | Parameter Name | Parameter Value |
|---|---|---|
| PSO-BLS | Iterations $N$ | 30 |
| | Population $P$ | 8 |
| | C1 | 4 |
| | C2 | 4 |
| | W | 0.9 |
| | Lower band $lb$ | [10 10 10] |
| | Upper band $ub$ | [100 100 100] |
| POA-BLS | Iterations $N$ | 30 |
| | Population $P$ | 8 |
| | Lower band $lb$ | [10 10 10] |
| | Upper band $ub$ | [100 100 100] |
| | $\Lambda$ | round (random (1,2)) |
| | $\gamma$ | 0.2 |
| IPOA-BLS | Iterations $N$ | 30 |
| | Population $P$ | 8 |
| | Lower band $lb$ | [10 10 10] |
| | Upper band $ub$ | [100 100 100] |
| | $\Lambda$ | $\theta$ (dynamic weight factor) |
| | $\gamma$ | 0.2 |

## 4. Experimental Results and Analysis

In order to state the advantages of the IPOA-BLS model in fault diagnosis, the classification results were compared with SVM, DBN, BLS, PSO-BLS, and POA-BLS diagnostic models. The classification results of each model are shown in Figure 6. In Figure 6, the blue circle (○) and red star (✱) mean the actual and predicted classification of the testing set samples, respectively. The blue circles and red stars overlap each other when the actual and predicted values of the classification are equal. If they are different, the blue circle and the red star do not overlap. The more overlapping means the fault resolution of the model is higher. In Figure 6a, the blue circle and red star overlap less, and the classification effect of the SVM model plays worse. In Figure 6f, the red star and the blue circle overlap each other the most times, which indicates that the IPOA-BLS model has the best classification effect.

To further visually reflect the fault identification capabilities of the model, the confusion matrix was used to visualize the fault recognition ability of SVM, DBN, BLS, PSO-BLS, POA-BLS, and IPOA-BLS models. The confusion matrix of each diagnostic model was calculated and plotted separately, and the results are shown in Figure 7. The diagonal elements of the confusion matrix indicate the number of samples that can be accurately classified, and the greater the number of diagonal elements, the better the classification performance of the model. The off-diagonal elements of the confusion matrix represent the number of samples that can be wrongly classified, and the smaller number of off-diagonal elements, the better the classification efficiency of the model.

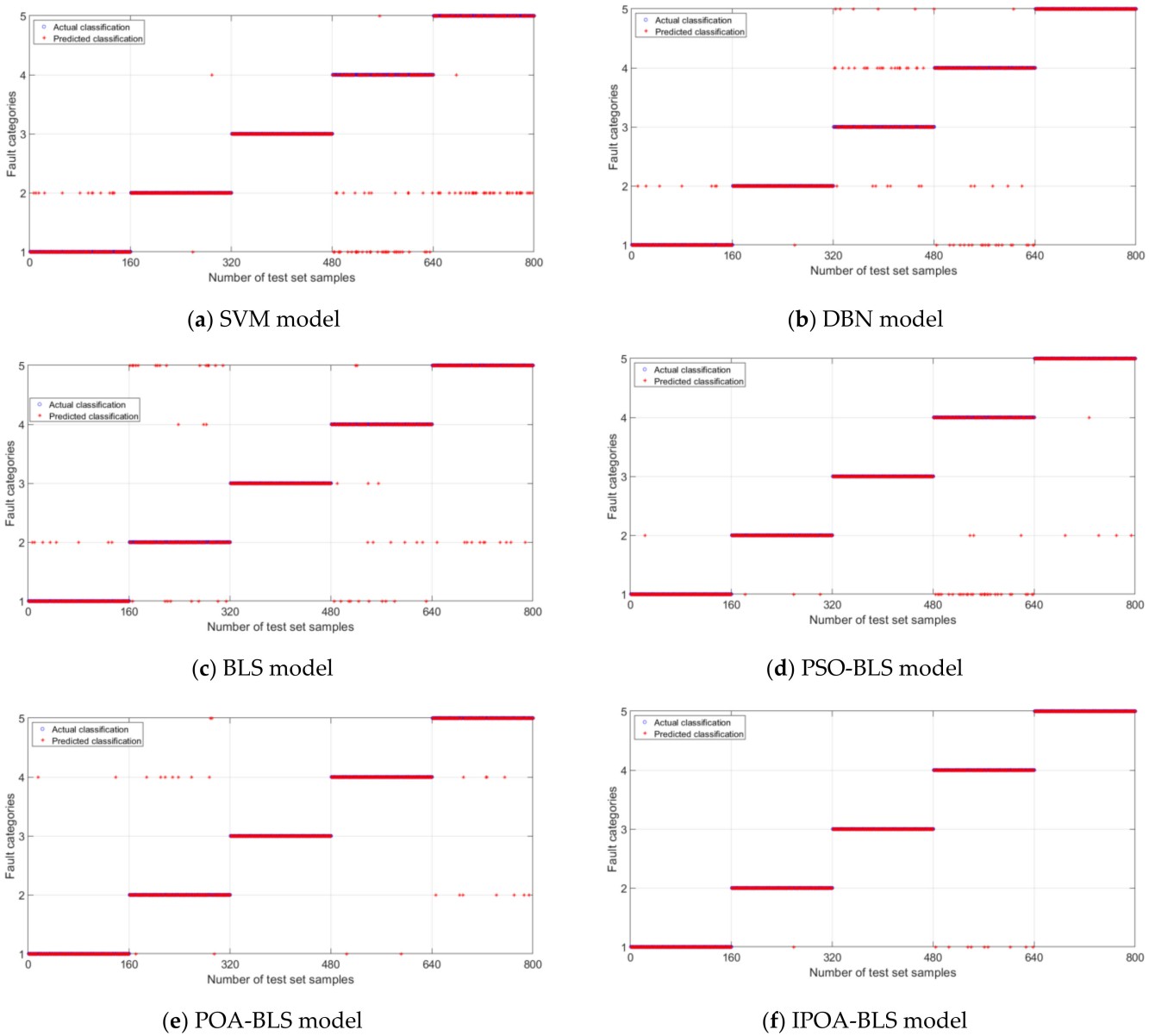

**Figure 6.** Classification result of the different models.

In Figure 7a, the SVM model can only accurately classify fault-F3. In total, 14 samples originally belonging to fault-F1 were misclassified as fault-F2. Two samples originally belonged to fault-F2; however, one of them was misclassified as fault-F1, and another one was misclassified as fault-F4, respectively. Altogether, 45 samples originally belonged to fault-F4; however, 1 of them was misclassified as fault-F5, 12 of them were misclassified as fault-F1, and 32 of them were misclassified as fault-F1, respectively. In total, 37 samples originally belonged to fault-F5, but 1 of them was misclassified as fault-F4, and 36 of them were misclassified as fault-F2, respectively. In Figure 7f, the IPOA-BLS model almost accurately classified fault-F1, fault-F2, fault-F3, and fault-F5. Only nine samples originally belonging to fault-F4 were misclassified as fault-F1.

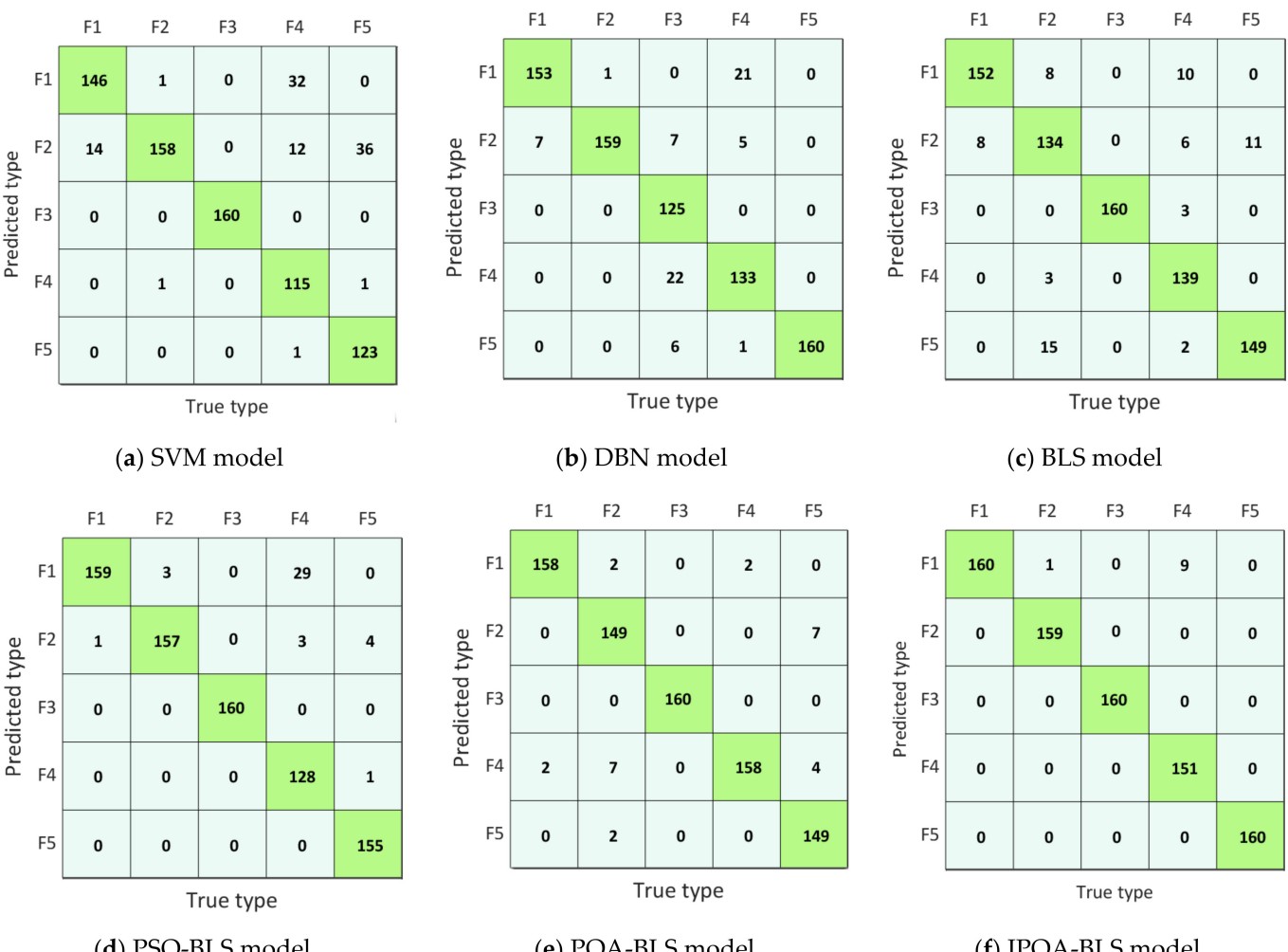

**Figure 7.** Confusion matrices of the different model.

It can be seen from Figure 7 that among the 800 test samples, the SVM model accurately classified 702 samples with an accuracy of 87.75%. The DBN model accurately classified 730 samples, and its accuracy was 91.25%. The BLS model exactly categorized 734 samples with an accuracy of 91.75%. The PSO-BLS model accurately classified 759 samples with an accuracy of 94.875%. The POA-BLS model rightly classified 774 samples, and its accuracy was 96.75%. The IPOA-BLS model exactly classified 790 samples with an accuracy of 98.75%. An observation can be made that the IPOA-BLS model was the most accurate compared with other models.

In order to better assess the performance of the proposed classification model, accuracy, precision, recall, and F1-score were used in the present study. The specific expressions are, respectively, as follows.

(1) The accuracy represents the number of correctly predicted samples as a percentage of the total number of samples.

$$accuracy = \frac{number\ of\ correctly\ classified\ samples}{total\ number\ of\ samples} \tag{12}$$

(2) The precision reflects the proportion of samples with a positive prediction that are really positive.

$$precision = \frac{TP}{TP + FP} \tag{13}$$

(3) The recall is the proportion of the number of correctly classified positive samples to the number of true positive samples.

$$recall = \frac{TP}{TP + FN} \tag{14}$$

(4) The *F*1-score is the weighted harmonic mean of precision and recall.

$$F1\text{-}score = \frac{2 \cdot precision \cdot recall}{precison + recall} \tag{15}$$

In the above equations, *TP* represents true positives; *FP* is false positives; and *FN* is false negatives.

In each fault, the classification performance metrics are shown in Table 9. It can be seen from Table 9 that the proposed model had the higher evaluation indicators no matter which evaluation standard was used.

**Table 9.** Performance indicators of fault classification model.

| Classification Model | Evaluation Index | Fault Types | | | | |
|---|---|---|---|---|---|---|
| | | F1 | F2 | F3 | F4 | F5 |
| SVM | Precision | 81.56% | 71.82% | 100% | 98.29% | 99.19% |
| | Recall | 91.25% | 98.75% | 100% | 71.88% | 76.88% |
| | F1-score | 86.14% | 83.16% | 100% | 83.03% | 86.62% |
| | Accuracy | 87.75% | | | | |
| DBN | Precision | 87.43% | 89.33% | 100% | 85.81% | 95.81% |
| | Recall | 95.63% | 99.38% | 78.12% | 83.13% | 100% |
| | F1-score | 91.34% | 94.08% | 87.72% | 84.44% | 97.86% |
| | Accuracy | 91.25% | | | | |
| BLS | Precision | 89.41% | 84.28% | 98.16% | 97.89% | 89.76% |
| | Recall | 95.00% | 83.75% | 100% | 86.88% | 93.13% |
| | F1-score | 92.12% | 84.04% | 99.07% | 92.05% | 91.41% |
| | Accuracy | 91.75% | | | | |
| PSO-BLS | Precision | 83.25% | 95.15% | 100% | 99.22% | 100% |
| | Recall | 99.38% | 98.12% | 100% | 80.00% | 96.88% |
| | F1-score | 90.60% | 96.62% | 100% | 88.58% | 98.41% |
| | Accuracy | 94.87% | | | | |
| POA-BLS | Precision | 97.53% | 95.51% | 100% | 92.40% | 98.68% |
| | Recall | 98.75% | 93.13% | 100% | 98.75% | 9313% |
| | F1-score | 98.14% | 94.30% | 100% | 95.47% | 95.82% |
| | Accuracy | 96.75% | | | | |
| IPOA-BLS | Precision | 94.14% | 100% | 100% | 100% | 100% |
| | Recall | 100% | 99.38% | 100% | 94.37% | 100% |
| | F1-score | 96.97% | 99.69% | 100% | 97.11% | 100% |
| | Accuracy | 98.75% | | | | |

The IPOA-BLS model has many advantages. Compared with the other models, there was a significant improvement in the classification accuracy in terms of WT fault diagnosis. The PCA method played an important role in dimension reduction and feature extraction,

and it was used to reduce the number of features of the experimental SCADA data from 52 to 35, which helped to decrease the calculation time of the model. The IPOA algorithm was evaluated through some different benchmark functions and was applied to the parameter's estimation of BLS. As can be seen from Figures 6 and 7, compared with the standard pelican optimization algorithm, the improved pelican optimization algorithm had a positive effect on BLS's parameters optimization. After optimizing the related parameters of BLS with the improved pelican optimization algorithm, the number of rightly classified samples increased by about 16, and the accuracy improved by 2%.

## 5. Conclusions

In this paper, in addition to the chaotic map strategy, the dynamic weight factor strategy was also used to improve the location of the standard pelican optimization algorithm in the initialization and moving towards prey stage; this helps to enhance the exploration of the improved pelican optimization algorithm technique. To improve the classification accuracy of the broad learning system model, the experimental data were normalized and the improved pelican optimization algorithm was used to optimize hyperparameters such as the number of feature nodes $N_f$, number of enhancement nodes $N_e$, and number of mapped feature layers $N_{fl}$. Performance indexes, classification accuracy, a classification results diagram, and confusion matrices fully demonstrate the excellent performance of the proposed classification model and verify the classification effect. The proposed model is very suitable for wind turbine fault classification in future practical applications.

The classification accuracy rate of the proposed model reaches a high level for typical wind turbine faults. Consequently, this model possesses great potential in wind turbine fault detection. In future studies, the development of the algorithms for fault feature extraction purposes will be investigated, and classification models based on deep learning will be analyzed.

**Author Contributions:** Data curation, M.H., L.G., Z.C.; methodology, C.X.; investigation and writing, W.T. All authors have read and agreed to the published version of the manuscript.

**Funding:** This research was funded by the Ministry of Science and Technology of Peoples Republic of China (Grant No. 2019YFE0104800), the Special Training Plan for Minority Science and Technology Talents of Natural Science Foundation of Xinjiang Uyghur Autonomous Region (Grant No. 2020D03004), the Fundamental Research Funds for the Central Universities (Grant No. B210201018), and the National Natural Science Foundation of China (Grant No. 52106238).

**Institutional Review Board Statement:** Not applicable.

**Informed Consent Statement:** Not applicable.

**Data Availability Statement:** The data that support this manuscript are available from Chang Xu upon reasonable request.

**Conflicts of Interest:** The authors declare no conflict of interest.

## Nomenclature

| | | | |
|---|---|---|---|
| AI | Artificial intelligence | GWO | Grey wolf optimizer |
| ANN | Artificial neural network | KNN | K-nearest neighbor |
| Average | Average value of the algorithm | MF | Mapped feature |
| Best | Best value of the algorithm | ML | Machine learning |
| BLS | Broad learning system | PCA | Principal component analysis |
| CNN | Convolutional neural networks | POA | Pelican optimization algorithm |
| DBN | Deep belief network | PSO | Particle swarm optimization |
| DL | Deep learning | RF | Random forest |
| EN | Enhancement nodes | RNN | Recurrent neural networks |

| F1 | Fault 1: the gearbox overheated | SAE | Stacked automatic encoders |
|---|---|---|---|
| F2 | Fault 2: the yaw system motor fault | Std. | Standard deviation of the algorithm |
| F3 | Fault 3: generator overheated | SVM | Support vector machine |
| F4 | Fault 4: invertor system fault | WOA | Whale optimization algorithm |
| F5 | Fault 5: pitch system fault | Worst | Worst value of the algorithm |
| IPOA | Improved pelican optimization algorithm | WT | Wind turbine |
| SCADA | Supervisory control and data acquisition | WWEA | World wind energy association |
| RVFLNN | Random vector functional-link neural network | | |
| PSO-BLS | Broad learning system model optimized by particle swarm optimization | | |
| POA-BLS | Broad learning system model optimized by pelican optimization algorithm | | |
| IPOA-BLS | Broad learning system model optimized by improved pelican optimization algorithm | | |

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
