# Peer review of "A Wind Turbine Fault Classification Model Using Broad Learning System Optimized by Improved Pelican Optimization Algorithm"

_machines, doi:10.3390/machines10050407_

Round 1

Reviewer 1 Report

The authors have presented an IPOA-BLS approach for wind turbine fault detection using SCADA data. Its an interesting work. The following comments need to be addressed for improving the manuscript. 

1) Broad learning system or board learning system? In title it is mentioned as Board. 

2) What is meant by broad learning system hyperparameters?  How it affects the performance. In abstract it need to be explained before introducing these terminologies. Similarly in appropriate places in the manuscript. 

3) Section 3: Heading needs grammatical correction. 

4) Wind turbine dataset description is missing. Need to explain in detail about the different conditions of the data used. Plot some data samples for understanding about the data. 

5) Conclusion need to be distinct from the manuscript. Therefore, don't abbreviate the terms in the first instance itself. The abbreviation should be written in expansion at least once, and it can be abbreviated subsequently while repeating the terms.  

6) Need more intuitive discussions on the experiments and results. 

Author Response

Responds to Reviewer’s Comments

Dear Reviewer:

Thank you for your comments concerning our manuscript entitled “A Wind Turbine Fault Classification Model using Broad Learning System Optimized by Improved Pelican Optimization Algorithm”. Those comments are all valuable and very helpful for revising and improving the guiding significance of our research. We have studied the comments carefully and have made some corrections which we hope to meet with approval. Revised portions are marked in red on the paper. The main correction in the paper and the responses to your comments are as follows:

Point 1: Broad learning system or board learning system? In the title, it is mentioned as Board.

Response 1: First of all, I am very sorry about my spelling mistakes, and thank you for your advice. The correct spelling of the word " Broad" in the Broad Learning System is "Broad", not "Board", and in this revision, we revised it in the manuscript.

Point 2: What is meant by broad learning system hyperparameters?  How it affects the performance. In abstract it needs to be explained before introducing these terminologies. Similarly, in appropriate places in the manuscript.

Response 2: Thank you very much for your suggestion. The Broad learning system (BLS) is also the same as other neural network models, which requires relevant parameters value to be set before use, and the parameter's value has a greater impact on the classification accuracy of the model. To address the problem that the settings of BLS hyperparameters affect the diagnosis result, In this paper, the hyperparameters of the BLS such as the number of feature nodes , number of enhancement nodes , and number of mapped features layers  were optimized by the improved pelican optimization algorithm(IPOA). In this revision, we added information about the hyperparameters of the BLS in the abstract section.

Point 3: Heading needs grammatical correction.

Response 3: Thank you very much for your advice. In this revision, we wrote the original title as “Wind turbine fault classification”.

Point 4: Wind turbine dataset description is missing. Need to explain in detail about the different conditions of the data used. Plot some data samples for understanding about the data.

Response 4: Thank you very much for your suggestion. In this revision, we revised the relevant contents (use red color to mark them) of this paper.

In this study, the information from the SCADA system recorded from 38 Vestas 1.5MW wind turbines in a wind farm in Liu He from 1 January 2018 to 1 December 2018 was used as experimental data and the time resolution of the data was ten minutes. After data cleaning, we coded the fault categories in the experimental data set such as the gearbox overheated (F1), yaw system motor fault (F2), generator overheated (F3), and inverter system fault (F4), and pitch system fault (F5). 800 samples were selected for each fault, of which 640 were used for training and 160 were used for testing. A total of 3200 samples were selected for training and 800 samples for testing.

In this experimental data, there are 52 observation characteristic parameters in of Vestas wind turbine, after feature extraction with principal component analysis (PCA) and comparison with Pearson correlation coefficient values we chose 35 characteristic parameters as the observation parameters of fault diagnosis. The relevant feature codes and their description is specified in Table 6. The feature code and its Pearson correlation coefficient of each fault are shown in Table 7.

Point 5: Conclusion needs to be distinct from the manuscript. Therefore, don't abbreviate the terms in the first instance itself. The abbreviation should be written in expansion at least once, and it can be abbreviated subsequently while repeating the terms.

Response 5: Thank you very much for your advice. According to your comments, in this revision, we rewrote the conclusion section (use red color to mark them) of this paper.

Point 6: Need more intuitive discussions on the experiments and results.

Response 6: Thank you very much for your suggestion. According to your comments, we have reworked and written the relevant content (use different colors to mark them) of this paper.

We appreciate your warm work earnestly and hope that the correction will meet with approval. Once again, thank you very much for your comments and suggestions.

Best regards.

Yours

sincerely.

Wumaier Tuerxun et al.

Reviewer 2 Report

The manuscript entitled “A Wind Turbine Fault Classification Model using Board Learning System Optimized by Improved Pelican Optimization Algorithm” deals with a very interesting and timely topic in wind energy research, which is the use of SCADA data for wind turbine fault diagnosis.

The manuscript is appropriate for the scientific objectives of the Machines journal.

The organization of the manuscript is appropriate, because the authors explain the methods and apply them to a test case study, by comparing against several other benchmark models widely used for SCADA data treatment in wind turbine fault diagnosis.

Nevertheless, in my opinion there are several aspects of this study which should be improved.

First of all, the English language should be carefully crosschecked by a native speaker. For example, in Table 6, the authors have written “peach system” instead of “pitch system”: errors like these are unacceptable in a scientific paper.

The Figures should be rearranged and their quality improved. For example, Figure 4 is practically impossible to read.

The employed data set should be described in much more detail. What is the averaging time of the data? What type of wind turbines are you dealing with? Most of all, how do the authors know what measurements correspond to faulty operation? My personal opinion is that a highest added value would be achieved by detecting faults without having at disposal information about the classification of samples, but I understand the rationale of the work done by the authors. Nevertheless, in my opinion it is fundamental to disclose how the authors have classified the samples as faulty or healthy.

As regards step 2 at page 12, how have the authors selected that 35 is the most appropriate number of features upon PCA reduction? This step of the algorithm seems quite arbitrary and should be explained better.

The Introduction section should be improved by posing better the proposed work in relation to the state of the art in the SCADA data analysis literature. I suggest including, for example, the following reference related to the use of PCA for dimensionality reduction and feature extraction:

Pozo, F., Vidal, Y., & Salgado, Ó. (2018). Wind turbine condition monitoring strategy through multiway PCA and multivariate inference. Energies, 11(4), 749.

Castellani, F., Astolfi, D., & Natili, F. (2021). SCADA data analysis methods for diagnosis of electrical faults to wind turbine generators. Applied Sciences, 11(8), 3307.

Wang, Y., Ma, X., & Qian, P. (2018). Wind turbine fault detection and identification through PCA-based optimal variable selection. IEEE Transactions on Sustainable Energy, 9(4), 1627-1635.

Author Response

Responds to Reviewer’s Comments

Dear Reviewer:

Thank you for your comments concerning our manuscript entitled “A Wind Turbine Fault Classification Model using Broad Learning System Optimized by Improved Pelican Optimization Algorithm”. Those comments are all valuable and very helpful for revising and improving the guiding significance of our research. We have studied the comments carefully and have made some corrections which we hope to meet with approval. Revised portions are marked in red on the paper. The main correction in the paper and the responses to your comments are as follows:

Point 1: First of all, the English language should be carefully crosschecked by a native speaker. For example, in Table 6, the authors have written “peach system” instead of “pitch system”: errors like these are unacceptable in a scientific paper.

Response 1: First of all, I am very sorry about my spelling mistakes, and thank you for your advice. The correct spelling of the word " pitch" in the pitch system is "pitch", not "peach", and in this revision, we revised it in the manuscript.

Point 2: The Figures should be rearranged and their quality improved. For example, Figure 4 is practically impossible to read.

Response 2: Thank you very much for your suggestion. In this revision, we revised and redrew Figure 4.

Point 3: The employed data set should be described in much more detail. What is the averaging time of the data? What type of wind turbines are you dealing with? Most of all, how do the authors know what measurements correspond to faulty operation? My personal opinion is that a highest added value would be achieved by detecting faults without having at disposal information about the classification of samples, but I understand the rationale of the work done by the authors. Nevertheless, in my opinion it is fundamental to disclose how the authors have classified the samples as faulty or healthy.

Response 3: Thank you very much for your suggestion. in this paper, we studied the fault state of the wind turbine, didn't study the normal state, and all fault-related data comes from the SCADA system. In this revision, we made a more detailed explanation of the experimental data and revised the relevant contents (use red color to mark them) of this paper.

In this study, the information from the SCADA system recorded from 38 Vestas 1.5MW wind turbines

in a wind farm in Liu He from 1 January 2018 to 31 December 2018 was used as experimental data and

the time resolution of the data was ten minutes. After data cleaning, we coded the fault categories in the

experimental data set such as the gearbox overheated (F1), yaw system motor fault (F2), generator

overheated (F3), inverter system fault (F4), and pitch system fault (F5). 800 samples were selected

for each fault, of which 640 were used for training and 160 were used for testing. A total of 3200 samples

were selected for training and 800 samples for testing.

In this experimental data, there are 52 observation characteristic parameters in of Vestas wind turbine,

after feature extraction with principal component analysis (PCA) and comparison with Pearson

correlation coefficient values we chose 35 characteristic parameters as the observation parameters of

fault diagnosis. The relevant feature codes and their description is specified in Table 6. The feature code

and its Pearson correlation coefficient of each fault are shown in Table 7.

Point 4: As regards step 2 at page 12, how have the authors selected that 35 is the most appropriate number of features upon PCA reduction? This step of the algorithm seems quite arbitrary and should be explained better.

Response 4: Thank you very much for your suggestion. in this paper, we studied the fault state of wind turbines and all fault-related data comes from the SCADA system. In this study, the information from the SCADA system recorded from 38 Vestas 1.5MW wind turbines in a wind farm in Liu He from 1 January 2018 to 1 December 2018 was used as experimental data and the time resolution of the data was ten minutes. After data cleaning, we coded the fault categories in the experimental data set such as the gearbox overheated (F1), yaw system motor fault (F2), generator overheated (F3), and inverter system fault (F4), and pitch system fault (F5).

In this experimental data, there are 52 observation characteristic parameters in of Vestas wind turbine, after feature extraction with PCA and comparison with Pearson correlation coefficient values we chose 35 characteristic parameters as the observation parameters of fault diagnosis. The relevant feature codes and their description is specified in Table 1.

Table 1. Feature code and its description (same as Table 6 in the original draft)

feature code

feature description

feature code

feature description

X1

impeller speed (rpm)

X19

pitch bearings temperature (℃)

X2

generator active power(kVA)

X20

pitch oil pressure (mPa)

X3

average reactive power (kVA)

X21

pitch system motor speed(rpm)

X4

average blade angle (deg)

X22

nacelle temperature (℃)

X5

bias of blade (m)

X23

converter cabinet temperature (℃)

X6

blade temperature (℃)

X24

Converter cabinet fan temperature (℃)

X7

paddle motor temperature (℃)

X25

yaw bearing radial force (kN)

X8

paddle system temperature (℃)

X26

yaw drive temperature (℃)

X9

ambient temperature (℃)

X27

yaw bearing axial force (kN)

X10

generator stator winding temperature u (℃)

X28

average wind speed (m/s)

X11

generator stator winding temperature v (℃)

X29

yaw inverter cabinet temperature (℃)

X12

generator stator winding temperature w (℃)

X30

yaw bearings temperature (℃)

X13

stator temperature (℃)

X31

yaw control cabinet temperature (℃)

X14

rotor temperature (℃)

X32

Wind direction angle

X15

rotor speed (rpm)

X33

transformer temperature (℃)

X16

gearbox oil temperature (℃)

X34

generator torque (kN.m)

X17

gearbox bearing temperature (℃)

X35

tower temperature (℃)

X18

gearbox oil pressure (mPa)

In this paper, 5 types of faults were classified, 800 samples were selected for each fault, of which 640 were used for training and 160 were used for testing. A total of 3200 samples were selected for training and 800 samples for testing. The feature values of some samples after normalization are shown in Table 2.

Table 2. Feature value of samples (same as Table 7 in the original draft)

fault codes

sample numbers

feature codes

X1

X2

...

X34

X35

F1

1

0.1744

0.7142

...

0.5468

0.4945

2

0.4264

0.1020

...

0.2906

0.5678

...

...

...

...

...

...

800

0.3305

0.6510

...

0.3681

0.5177

F2

1

0.2411

0.6758

...

0.5318

0.2808

2

0.2636

0.6988

...

0.6312

0.5426

...

...

...

...

...

...

800

0.3140

0.7125

...

0.6095

0.3846

F3

1

0.3259

0.6169

...

0.2302

0.4881

2

0.3808

0.6096

...

0.3883

0.5971

...

...

...

...

...

...

800

0.6320

0.5055

...

0.0114

0.6165

F4

1

0.7550

0.6822

...

0.2370

0.3723

2

0.6868

0.6455

...

0.2577

0.3979

...

...

...

...

...

...

800

0.6367

0.6873

...

0.4328

0.5768

F5

1

0.4428

0.6877

...

2

0.8318

0.6664

...

...

...

...

...

...

...

800

0.8987

0.9026

...

0.4819

0.5360

The amount of features affecting each fault is not 35(see table 1), and the number of features for each fault is not the same. The feature code and its Pearson correlation coefficient of each fault are shown in Table 3.

Table 3. Feature code (Fc) and its Pearson correlation coefficient (Pcc) of each fault

Fc

Pcc

Fc

Pcc

Fc

Pcc

Fc

Pcc

Fc

Pcc

gearbox overheated (F1)

X2

0.3872

X9

0.1736

X10

0.4436

X11

0.4507

X12

0.4523

X13

0.3154

X14

0.3511

X15

0.4625

X16

0.7831

X17

0.8865

X18

0.3221

X22

0.1023

X28

0.4178

yaw system motor fault (F2)

X2

0.3661

X25

0.3622

X26

0.6134

X27

0.3531

X28

0.3164

X29

0.5171

X30

0.5731

X31

0.5463

X32

0.5273

generator overheated (F3)

X2

0.4175

X3

0.4336

X10

0.6734

X11

0.6804

X12

0.6779

X13

0.7863

X14

0.7162

X15

0.5153

X22

0.1227

X34

0.5271

converter system fault (F4)

X14

0.4267

X22

0.1103

X23

0.6832

X24

0.6054

X33

0.3762

X35

0.1272

pitch system fault (F5)

X1

0.3874

X2

0.3642

X25

0.4623

X26

0.6723

X27

0.4167

X28

0.3465

X29

0.5732

X30

0.6264

X31

0.6114

X32

0.4178

If we use the IPOA-BLS model for fault classification, according to the feature codes of each fault in Table 3, the corresponding feature code value in Table 2 is retained and the rest are zero. For example, the features that affect Fault 3 mainly include X2, X3, X10, X11, X12, X13, X14, X15, X22, and X34. The corresponding feature codes value in Table 2 is retained and the rest are zero, as follows:

X1

X2

X3

……

X10

X11

X12

X13

X14

X15

……

X22

…..

X34

X35

1

0

0.3259

0.4179

0

0.1449

0.0964

0.5181

0.2819

0.3432

0.8973

0

0.4490

0.2302

0

2

0

0.6096

0.3707

0

0.1925

0.0597

0.1849

0.5110

0.1644

0.8527

0

0.3615

0.3883

0

...

800

0

0.5055

0.4272

0

0.1374

0.0690

0.3783

0.7315

0.0838

0.7997

0

0.3735

0.2114

0

In this revision, we added Table 3 (same as Table 7 in the revised manuscript) , and deleted Table 2 (same as Table 7 in the original draft)

Point 5: The Introduction section should be improved by posing better the proposed work in relation to the state of the art in the SCADA data analysis literature. I suggest including, for example, the following reference related to the use of PCA for dimensionality reduction and feature extraction:

Pozo, F., Vidal, Y., & Salgado, Ó. (2018). Wind turbine condition monitoring strategy through multiway PCA and multivariate inference. Energies, 11(4), 749.

Castellani, F., Astolfi, D., & Natili, F. (2021). SCADA data analysis methods for diagnosis of electrical faults to wind turbine generators. Applied Sciences, 11(8), 3307.

Wang, Y., Ma, X., & Qian, P. (2018). Wind turbine fault detection and identification through PCA-based optimal variable selection. IEEE Transactions on Sustainable Energy, 9(4), 1627-1635.

Response 5: Thank you very much for your advice. In this revision, we cited and discussed the relevant references in the introduction section of this paper.

We appreciate your warm work earnestly and hope that the correction will meet with approval. Once again, thank you very much for your comments and suggestions.

Best regards.

Yours

sincerely.

Wumaier Tuerxun et al.

Reviewer 3 Report

The manuscript proposes and optimization method of broad learning system hyperparameters based on the improved pelican optimization algorithm (IPOA-BLS). The IPOA-BLS is illustrated to classify different failures of wind turbines (WT) and it is compared with different techniques to discuss the classification accuracy in terms of WT fault diagnosis. Therefore, this reviewer suggests that the manuscript cannot be published in the present for the reasons that follow.

Major issues

- Section 2.3. There is an improvement in convergence with IPOA, however, the IPOA method does not result with the lowest standard deviation in all benchmark functions and this parameter is important.

- More discussion is needed on Figure 12 as well as a further discussion of all the advantages that IPOA-BLS has over the other techniques.

Minor issues

-In the conclusions section. Make some specific and conclusive statements based on the results.

-The quality of figures 4 and 5 needs to be improved.

-It is necessary to define nomenclatures such as PSO, GWO, WOA, ITSO, among others, and their respective reference.

-More details are required on how to obtain the 3200 examples in Table 7.

-It is suggested to make a table on the observed in figures 6-11.

Author Response

Responds to Reviewer’s (#3) Comments

Dear Reviewer:

Thank you for your comments concerning our manuscript entitled “A Wind Turbine Fault Classification Model using Broad Learning System Optimized by Improved Pelican Optimization Algorithm”. Those comments are all valuable and very helpful for revising and improving the guiding significance of our research. We have studied the comments carefully and have made some corrections which we hope to meet with approval. Revised portions are marked in red on the paper. The main correction in the paper and the responses to your comments are as follows:

Point 1: Section 2.3. There is an improvement in convergence with IPOA, however, the IPOA method does not result with the lowest standard deviation in all benchmark functions and this parameter is important.

Response 1: Thank you very much for your suggestion. In this revision, in addition to the Tent chaotic map strategy, we also used the dynamic weight factor strategy. The statistical results were recalculated(Tabel 3-5), and the convergence curve was redrawn(Figure 4).

Point 2: More discussion is needed on Figure 12 as well as a further discussion of all the advantages that IPOA-BLS has over the other techniques.

Response 2: Thank you very much for your advice. In the first draft, the bar chart was drawn in this paper (Figure 12) contains more information about the performaces data, and difficult to show the exact value of the relevant performance indicators. To avoid this, in this revision, we revised the relevant contents and replaced the bar chart with a table (Table 9).

Point 3: In the conclusions section. Make some specific and conclusive statements based on the results.

Response 3: Thank you very much for your advice. In this revision, we have rewritten the conclusions section (use a different color to mark them) of this article.

Point 4: The quality of figures 4 and 5 need to be improved.

Response 4: Thank you very much for your suggestion. In the first draft, Figure 4 was really difficult to read. In this revision, we recalculated the statistical results (Tabel 3-5), and redrawn Figure 4.

Figure 5 represents the classification flow of the IPOA-BLS model. For intuition and ease to understand, we added relevant mini-figures (it is optional, and can also be deleted) in the classification result and evaluation index section of Figure 5, as follows:

mini figure

mini figure

Figure 5. classification process of the IPOA-BLS model

The detailed figure of mini-figures was shown in Figure 6, Figure 7, and Table 9, therefore, in this revision we didn't change Figure 5.

Point 5: It is necessary to define nomenclatures such as PSO, GWO, WOA, and IPOA, among others, and their rRespective reference.

Response 5: Thank you very much for your advice. More than 30 abbreviations were used in this paper. According to your comments and for ease of reading, in this revision, we added the nomenclature table of this paper (please check page 2).

Point 6: More details are required on how to obtain the 3200 examples in Table 7.

Response 6: Thank you very much for your suggestion. In this revision, we revised the relevant contents (use red color to mark them) of this paper.

In this study, the information from the SCADA system recorded from 38 Vestas 1.5MW wind turbines in a wind farm in Liu He from 1 January 2018 to 1 December 2018 was used as experimental data and the time resolution of the data was ten minutes. After data cleaning, we coded the fault categories in the experimental data set such as the gearbox overheated (F1), yaw system motor fault (F2), generator overheated (F3), and inverter system fault (F4), and pitch system fault (F5). 800 samples were selected for each fault, of which 640 were used for training and 160 were used for testing. A total of 3200 samples were selected for training and 800 samples for testing.

In this experimental data, there are 52 observation characteristic parameters in of Vestas wind turbine, after feature extraction with principal component analysis (PCA) and comparison with Pearson correlation coefficient values we chose 35 characteristic parameters as the observation parameters of fault diagnosis. The relevant feature codes and their description is specified in Table 6. The feature code and its Pearson correlation coefficient of each fault are shown in Table 7.

Point 7: It is suggested to make a table on the observed in figures 6-11.

Response 7: Thank you very much for your suggestion. In the first draft, we plotted the figures of classification results and confusion matrices for each model(Figure 6-11), and many sentences were reused in the interpretation. These reused sentences reduced the quality of the paper.

In order to improve the quality of the article, in this revision, we rewrote the relevant content. we explained the classification results and the confusion matrix part separately and tried to avoid the repetition of sentences.

We appreciate your warm work earnestly and hope that the correction will meet with approval. Once again, thank you very much for your comments and suggestions.

Best regards.

Yours

sincerely.

Wumaier Tuerxun et al.

Reviewer 4 Report

Manuscript Number: - machines-1682082-peer-review-v1 
Submitted to: MDPI machines 
Title: A Wind Turbine Fault Classification Model using Board Learning System Optimized by Improved Pelican Optimization Algorithm

Report of Review

This paper proposes an optimization method of BLS hyper-parameters based on the improved pelican optimization algorithm (IPOA), and build an IPOA-BLS fault classification model. The results show that compared with other models, IPOA-BLS may improve the accuracy of WT fault diagnosing.

Queries:

Revise the title and make it concise.
Add a list of symbols and acronyms, and define acronym at first place of use, for : what PCA, POA, IPO, ITSO , PSO, GWO, WOA, POA , IPOA,  SCADA etc.........
The Abstract should contain answers to the following questions: What problem was studied and why is it important? What methods were used? What are the important results? What conclusions can be drawn from the results? What is the novelty of the work and where does it go beyond previous efforts in the literature? 
The paper structure is not well organized. There is no coordination between sections and subsection and a lot of repetitions. Also, there are short and fragmented sentences without coordinates.
Figure 2 texts are too small to read.
Check eqt (3) some characters are strange.
Why the inverse is noted by a superscript (+) in eqt(4) it should be ( -1).
Explain why  is randomly equal to 1 or 2, why typical these two values only. For example the normal random number of normal distribution between [0,1]
Sentence line 173-174 unclear:    The problem of in coordination with local development performance.
 I can’t see how the formula (5) is rewritten as (9)
  In Table 2. Benchmark functions’ provide each function by its reference for more credibility.
In Table 3. What the second column stands for: Best, Worst, Average, Std. Please give comments.
In the graphical illustrations the labels are too small to read.
In lines 204 – 206 is unclear what are referring to for this two sample sizes.
In Table 8. Change:!  Lower board and Upper board to Lower band and Upper band  

Rewrite the whole paragraph lines 257-265, there is bad structure and none of full stops.
Rewrite the whole paragraph lines 267-272, there is bad structure and none of full stops.
Rewrite the whole paragraph lines 277-282, there is bad structure and none of full stops.
Rewrite the whole paragraph lines 284-289, there is bad structure and none of full stops.
Rewrite the whole paragraph lines 291-296, there is bad structure and none of full stops.
Rewrite the whole paragraph lines 298-304, there is bad structure and none of full stops.
Improve the discussions of figures from 6 to 11; what is written is only comments which sjould be an in-depth analysis.
Meaning of eqt 18-19-20 should be better described.
Line 310 figure 10 should be 11
In conclusion correct:
For the typical WT fault diagnosis, the IPOA-BLS classification accuracy rate could reach a high level, there is but still exists some problems, such as a difficulty to distinguish the complex faults. In order to further improve the classification accuracy of the model, a more  suitable further fault feature extraction methods need to be explored, it is also the next step for our 342 research direction.

 Conclusion should be reformulated. Focus briefly on the presented methodology, the findings and practical feasibility and the perspectives.

Author Response

Responds to Reviewer’s Comments

Dear Reviewer:

Thank you for your comments concerning our manuscript entitled “A Wind Turbine Fault Classification Model using Broad Learning System Optimized by Improved Pelican Optimization Algorithm”. Those comments are all valuable and very helpful for revising and improving the guiding significance of our research. We have studied the comments carefully and have made some corrections which we hope to meet with approval. Revised portions are marked in red on the paper. The main correction in the paper and the responses to your comments are as follows:

Point 1: Revise the title and make it concise.

Response 1: Thank you very much for your advice. In this paper, we use an improved pelican optimization algorithm (IPOA) to optimize the parameters of the broad learning system (BLS), such as the number of feature nodes , number of enhancement nodes , and number of mapped features layers , and then to construct the IPOA - BLS classification model. In order to fully reflect the core content of the paper, the title is written as “A Wind Turbine Fault Classification Model using Broad Learning System Optimized by Improved Pelican Optimization Algorithm”. If it is needed to be simplified, we think the title can be written as “A Wind Turbine Fault Classification Model using BLS Optimized by Improved Pelican Optimization Algorithm” or “A Wind Turbine Fault Classification Model using IPOA-BLS”. However, we still recommend not changing the title of the paper.

Point 2: Add a list of symbols and acronyms, and define acronym at first place of use, for : what PCA, POA, IPO, ITSO , PSO, GWO, WOA, POA , IPOA,  SCADA etc…

Response 2: Thank you very much for your advice. More than 30 abbreviations were used in this paper. According to your comments and for ease of reading, in this revision, we added the nomenclature table of this paper (please check page 2).

Point 3: The Abstract should contain answers to the following questions: What problem was studied and why is it important? What methods were used? What are the important results? What conclusions can be drawn from the results? What is the novelty of the work and where does it go beyond previous efforts in the literature?

Response 3: Thank you very much for your suggestion. According to your comments, in this revision, we rewrote the Abstract section of this article.

Point 4: The paper structure is not well organized. There is no coordination between sections and subsection and a lot of repetitions. Also, there are short and fragmented sentences without coordinates.

Response 4: Thank you very much for your suggestion. According to your comments, in this revision, we re-formatted and rewrote the relevant contents of the paper.

Point 5: Figure 2 texts are too small to read.

Response 5: Thank you very much for your suggestion. According to your comments, in this revision, we rescaled and enlarged Figure 2.

Point 6: Check eqt (3) some characters are strange.

Response 6: Thank you very much for your suggestion. According to your comments, in this revision, we revised the relevant formula.

Point 7: Why the inverse is noted by a superscript (+) in eqt(4) it should be ( -1).

Response 7: First of all, I am very sorry about my typing mistakes, and thank you very much for your suggestion. According to your comments, in this revision, we revised the relevant formula.

Point 8: Explain why is randomly equal to 1 or 2, why typical these two values only. For example the normal random number of normal distribution between [0,1].

Response 8: First of all, thank you very much for your suggestion. In the original article “Pelican Optimization Algorithm: A Novel Nature-Inspired Algorithm for Engineering Applications. Sensors-Basel 2022, 22, (3), 855. https://doi.org/10.3390/s22030855” when the authors explain Equation 4, said, is randomly equal to 1 or 2. In this revision, in addition to the Tent chaotic map strategy, we also used the dynamic weight factor strategy to improve the relevant formula( as below) in the standard POA algorithm. The parameter is not present in the revised formula of the improved pelican optimization algorithm (IPOA).

Formula

Source of paper

Initialization : Equation(1):

(Original standard POA algorithm)

Pelican Optimization Algorithm: A Novel Nature-Inspired Algorithm for Engineering Applications. Sensors-Basel 2022, 22, (3), 855.

 https://doi.org/10.3390/s22030855

Moving towards Prey: Equation(4):

Initialization strategy:(was used Tent chaotic map)

(Improved POA algorithm)

 In this paper, in this revision

if the initial population generated by the tent chaotic map is used, it helps to improve the global search performance of the POA algorithm.

Moving towards prey: (was used dynamic weight factor )

the dynamic weight factor helps the pelican to constantly update its position. At the beginning of the iteration has a large value, when the pelican can perform a better global search, and at the end of the iteration decreases adaptively, and this time the pelican can perform a better local search while increasing the convergence speed.

Point 9: Sentence line 173-174 unclear: The problem of in coordination with local development performance.

Response 9: Thank you very much for your suggestion. According to your comments, in this revision, we revised the relevant contents of the paper.

Point 10: I can’t see how the formula (5) is rewritten as (9).

Response 10: First of all, I am very sorry about my typing mistakes, and thank you very much for your suggestion. In this paper, formula(5) represents the position of the pelicans in the dimensional space. formula(6) and formula (9) represent the position of the i-th pelican in the m-th dimension. The structure of formula 6 and formula 9 is different, please see the following table

Formula 5:

Standard pelican optimization algorithm (POA)

Formula 6:

In the initialization stage, the pelican's position is randomly distributed.

Standard pelican optimization algorithm (POA)

Formula 9:

In the initialization stage, the pelican’s position is distributed according to the Tent chaotic map method.

Improved pelican optimization algorithm (IPOA)

Point 11: In Table 2. Benchmark functions’ provide each function by its reference for more credibility.

Response 11: Thank you very much for your suggestion. According to your comments, in this revision, we revised the relevant contents of the paper.

Point 12: In Table 3. What the second column stands for: Best, Worst, Average, Std. Please give comments.

Response 12: Thank you very much for your advice. In Table 3-5, “Best” represents the best value of the algorithm, “Worst” denotes the worst value of the algorithm, “Average” denotes the Average value of the algorithm, and “Std.” denotes the Standard deviation of the algorithm. According to your comments, in this revision, we revised the relevant contents of the paper.

Point 13: In the graphical illustrations the labels are too small to read.

Response 13: Thank you very much for your suggestion. In this revision, we revised and redrew relevant figures.

Point 14: In lines, 204 – 206 is unclear what are referring to for these two sample sizes.

Response 14: Thank you very much for your suggestion. In this paper, 5 types of faults were classified, 800 samples were selected for each fault, of which 640 were used for training and 160 were used for testing. A total of 3200 samples were selected for training and 800 samples for testing. Please check the following table.

fault code

number of a training set

number of a testing set

total

fault 1

640

160

800

fault 2

640

160

800

fault 3

640

160

800

fault 4

640

160

800

fault 5

640

160

800

total

3200

800

4000

Point 15: In Table 8. Change:!  Lower board and Upper board to Lower band and Upper band.

Response 15: Thank you very much for your advice. According to your comments, in this revision, we revised the relevant contents of the paper.

Point 16: Rewrite the whole paragraph lines 257-265, there is bad structure and none of full stops.

Rewrite the whole paragraph lines 267-272, there is bad structure and none of full stops.

Rewrite the whole paragraph lines 277-282, there is bad structure and none of full stops.

Rewrite the whole paragraph lines 284-289, there is bad structure and none of full stops.

Rewrite the whole paragraph lines 291-296, there is bad structure and none of full stops.

Rewrite the whole paragraph lines 298-304, there is bad structure and none of full stops..

Response 16: Thank you very much for your suggestion. In the first draft, we plotted the figures of classification results and confusion matrices for each model(Figure 6-11), and many sentences were reused in the interpretation. These reused sentences reduced the quality of the paper.

In order to improve the quality of the article, in this revision, we rewrote the relevant content. we explained the classification results and the confusion matrix part separately and tried to avoid the repetition of sentences.

Point 17: Improve the discussions of figures from 6 to 11; what is written is only comments which should be an in-depth analysis.

Response 17: Thank you very much for your advice. According to your comments, we rewrote the result and discussion section (use red color to mark them) of this paper.

Point 18: Line 310 figure 10 should be 11

Response 18: Thank you very much for your advice. According to your comments, in this revision, we revised the relevant contents of the paper.

Point 19: In conclusion correct: “For the typical WT fault diagnosis, the IPOA-BLS classification accuracy rate could reach a high level, there is but still exists some problems, such as a difficulty to distinguish the complex faults. In order to further improve the classification accuracy of the model, a more suitable further fault feature extraction methods need to be explored, it is also the next step for our research direction.” Conclusion should be reformulated. Focus briefly on the presented methodology, the findings and practical feasibility and the perspectives.

Response 19: Thank you very much for your advice. According to your comments, we rewrote the conclusion section (use red color to mark them) of this paper.

We appreciate your warm work earnestly and hope that the correction will meet with approval. Once again, thank you very much for your comments and suggestions.

Best regards.

Yors

sincerely.

Wumaier Tuerxun et al.

Round 2

Reviewer 2 Report

The manuscript has been considerably improved and I sincerely appreciate the effort done by the authors.

An important point should be clarified better. In relation to Response 3, the authors claim that "all fault-related data comes from the SCADA system".

What does this mean? How do the authors know the data they have selected correspond to a fault? Have the authors employed a further information source, as for example the alarm logs of the wind turbines? Or have the authors inferred this from the SCADA data themselves? If this is the case, and they already know through SCADA data analysis that there is a fault, what is exactly the usefulness of their method? Recognizing what they already knew? Therefore, the relevance of the method depends on how the authors have selected the faulty data and basing on what information. For this reason, the explanation can not be vague.

Author Response

Responds to Reviewer’s (#2) Comments

Dear Reviewer:

Thank you for your comments concerning our manuscript entitled “A Wind Turbine Fault Classification Model using Broad Learning System Optimized by Improved Pelican Optimization Algorithm”. Those comments are all valuable and very helpful for revising and improving the guiding significance of our research. We have studied the comments carefully and have made some corrections which we hope to meet with approval. The main correction in the paper and the responses to your comments are as follows:

Point 1: What does this mean? How do the authors know the data they have selected correspond to a fault? Have the authors employed a further information source, as for example the alarm logs of the wind turbines? Or have the authors inferred this from the SCADA data themselves? If this is the case, and they already know through SCADA data analysis that there is a fault, what is exactly the usefulness of their method? Recognizing what they already knew? Therefore, the relevance of the method depends on how the authors have selected the faulty data and basing on what information. For this reason, the explanation can not be vague.

Response 1: Thank you very much for your suggestion. in this paper, we studied the fault state of the wind turbine, didn't study the normal state, and all fault-related data comes from the SCADA system with alarming logs of the wind turbines.
At present, most of wind turbines are equipped with SCADA system, which obtains wind turbine operation data through various sensors. It can make the online monitoring of wind turbine operating indexes, working environment, grid connection and other parameters, and also to alarm and record the fault signals monitored, so as to facilitate the checking and maintenance of failure parts and improve the safety and stability of the wind turbine operation.
Most of the alarms of SCADA system still uses the alarm method of upper and lower limit values of relevant parameters, and the alarm threshold value is usually set according to the range of each detected parameter during normal operation of the unit, and different types of parameters have different threshold values. For example, the normal temperature range of the wind turbine gearbox oil temperature is (20℃ ,75℃). If the relevant sensor measures that the current oil temperature is 78℃, the SCADA system will alarm that the gearbox oil temperature has increased.
The SCADA system only alerts fault based on thresholds, and relying on threshold alarms is a very poor way to discriminate.
The failure of the SCADA system alarmed is only a preliminary diagnosis, and needs to be further analyzed.
Although the SCADA system has an alarm function, it still relies on poor discrimination method. SCADA data has a lot of abnormal noise data, repeated and missing values. We use effective methods to analyze and pre-process these data, select reliable, valuable fault data, and then to classify them. By doing this, faults are detected and identified in advance.

We appreciate your warm work earnestly and hope that the correction will meet with approval. Once again, thank you very much for your comments and suggestions.

Best regards.
Yours
sincerely.
Wumaier Tuerxun et al.

Reviewer 3 Report

The manuscript proposes and optimization method of broad learning system hyperparameters based on the improved pelican optimization algorithm (IPOA-BLS). The IPOA-BLS is illustrated to classify different failures of wind turbines (WT) and it is compared with different techniques such as support vector machine, deep belief network, broad learning system, broad learning system model optimized by particle swarm optimization and broad learning system model optimized by pelican optimization algorithm. All techniques are discussed on the classification accuracy in terms of WT fault diagnosis. The manuscript has been improved and this reviewer suggests that the article can be published.

Author Response

Responds to Reviewer’s (#3) Comments

Dear Reviewer:

Thank you for your comments concerning our manuscript entitled “A Wind Turbine Fault Classification Model using Broad Learning System Optimized by Improved Pelican Optimization Algorithm”. Those comments are all valuable and very helpful for revising and improving the guiding significance of our research. We have studied the comments carefully and have made some corrections which we hope to meet with approval. Revised portions are marked in red on the paper. The main correction in the paper and the responses to your comments are as follows:

Point 1: The manuscript proposes and optimization method of broad learning system hyperparameters based on the improved pelican optimization algorithm (IPOA-BLS). The IPOA-BLS is illustrated to classify different failures of wind turbines (WT) and it is compared with different techniques such as support vector machine, deep belief network, broad learning system, broad learning system model optimized by particle swarm optimization and broad learning system model optimized by pelican optimization algorithm. All techniques are discussed on the classification accuracy in terms of WT fault diagnosis. The manuscript has been improved and this reviewer suggests that the article can be published.

Response 1: First of all, Thank you very much for your omments. Without your comments, the quality of the paper would not have improved. After we revised the paper according to your suggestions, the quality of the paper has been further improved.

We appreciate your warm work earnestly and hope that the correction will meet with approval. Once again, thank you very much for your comments and suggestions.

Best regards.

Yours

sincerely.

Wumaier Tuerxun et al.

Reviewer 4 Report

Manuscript Number: - machines-1682082-peer-review-v2

Submitted to: MDPI machines

Title:  A Wind Turbine Fault Classification Model using Broad Learning System Optimized by Improved Pelican Optimization Algorithm

Report of Review

This paper still requires improved editing and further clarification in order to be accepted.

Please show how the answers of the past and present queries are reflected in the paper text.

Queries:

  1. Please check again the paper language; still there are style and grammatical errors.
  2. If an acronym is defined at its first place of use, do not repeat again its definition. I have noticed such repetition in the text and notably in the conclusion.
  3. Please do not use “ we” better to use indefinite
  4. The Abstract still needs to be reformulated; it is of bad style
  5.  In Table 2. Benchmark functions’ provide each function by its reference for more credibility.
  6. In fig.5 the labels are too small to read.
  7. Meaning of equations 12-15 should be better described.
  8. Conclusion should be reformulated. Focus briefly on the presented methodology, the findings and practical feasibility and the perspectives.

Author Response

Responds to Reviewer’s (#4) Comments

Dear Reviewer:

Thank you for your comments concerning our manuscript entitled “A Wind Turbine Fault Classification Model using Broad Learning System Optimized by Improved Pelican Optimization Algorithm”. Those comments are all valuable and very helpful for revising and improving the guiding significance of our research. We have studied the comments carefully and have made some corrections which we hope to meet with approval. The main correction in the paper and the responses to your comments are as follows:

Point 1: Please check again the paper language; still there are style and grammatical errors.

Response 1: Thank you very much for your suggestion. According to your comments, in this revision, we revised grammatical errors of the relevant contents.

Point 2: If an acronym is defined at its first place of use, do not repeat again its definition. I have noticed such repetition in the text and notably in the conclusion.

Response 2: Thank you very much for your advice. According to your comments, in this revision, we revised the relevant contents of the paper.

Point 3: Please do not use “ we” better to use indefinite

Response 3: Thank you very much for your advice. According to your comments, in this revision, we revised the relevant contents of the paper, and didn’t use “we”.

Point 4: The Abstract still needs to be reformulated; it is of bad style.

Response 4: Thank you very much for your advice. According to your comments, in this revision, we rewrote the Abstract section of this article. The specific structure of the Abstract section is as follows:

(1) As a classification model, broad learning system is widely used in wind turbine fault diagnosis. However, the setting of hyperparameters for the models directly affects the classification accuracy of the models and it generally relies on practical experience and prior knowledge. (answered, What problem was studied and why is it important? )

(2) In order to effectively solve the above problems, the parameters of the broad learning system such as the number of feature nodes, the number of enhancement nodes, and the number of mapped features layer were optimized by the improved pelican optimization algorithm, and was build a classification model based on broad learning system optimized by the improved pelican optimization algorithm. (answered, What methods were used? )

(3) The classification accuracy of the proposed model was the highest, reached 98.75%. (answered, What are the important results? )

(4) It is further shown that compared with the support vector machine, deep belief networks, and broad learning system models optimized by particle swarm optimization algorithm, the proposed model effectively improves the accuracy of wind turbine fault diagnosing. (answered, What conclusions

can be drawn from the results? What is the novelty of the work and where does it go beyond previous

efforts in the literature? )

Point 5: In Table 2. Benchmark functions’ provide each function by its reference for more credibility.

Response 5: Thank you very much for your suggestion. According to your comments, in this revision, we added the optimal reference value of the benchmark functions (was revised according to my understanding).

Description of the benchmark functions are shown in Table 2. In this Table, f 1-f 4 are the unimodal test functions, f 5-f 7 are the multimodal test functions, f 8-f 10 are the composite benchmark functions.

Point 6: In fig.5 the labels are too small to read.

Response 6: Thank you very much for your suggestion. Figure 5 represents the classification flow of the IPOA-BLS model. For intuition and ease to understand, we added relevant mini-figures (it is optional, and can also be deleted) in the classification result and evaluation index section of Figure 5, as follows:

mini figure

mini figure

Figure 5. classification process of the IPOA-BLS model

The detailed figure of mini-figures was shown in Figure 6, Figure 7, and Table 9.

Point 7: Meaning of equations 12-15 should be better described.

Response 7: Thank you very much for your suggestion. According to your comments, in this revision, we revised the relevant formula.

Point 8: Conclusion should be reformulated. Focus briefly on the presented methodology, the findings and practical feasibility and the perspectives.

Response 8: Thank you very much for your advice. According to your comments, in this revision, we rewrote the Conclusion section of this article. The specific structure of the Conclusion section is as follows:

(1) In this paper, in addition to the chaotic map strategy, the dynamic weight factor strategy was also used to improve the location of the standard pelican optimization algorithm in the initialization and moving towards the prey stage, this helps to enhance the exploration of the improved pelican optimization algorithm technique. (presented methodology 1--- to improve standard POA)

(2)To improve the classification accuracy of the broad learning system model, the experimental data were normalized and the improved pelican optimization algorithm was used to optimize its hyperparameters such as number of feature nodes, number of enhancement nodes, and number of mapped feature layers. (presented methodology 2--- to optimize parameters of BLS)

(3)Through performance indexes, classification accuracy, classification results diagram, and confusion matrices fully demonstrate the excellent performance of the proposed classification model and verify the classification effect. The proposed model is very suitable for wind turbine fault classification in future practical applications. (findings and practical feasibility)

(4)The classification accuracy rate of the proposed model reaches a high level for typical wind turbine faults. Consequently, this model possesses great potential in wind turbine fault detection. In future studies, the development of the algorithms for fault feature extraction purposes will be investigated, and classification models based on deep learning will be analyzed. (future work)

We appreciate your warm work earnestly and hope that the correction will meet with approval. Once again, thank you very much for your comments and suggestions.

Best regards.

Yours

sincerely.

Wumaier Tuerxun et al.

Round 3

Reviewer 2 Report

The paper can be accepted because the explanation provided by the authors is finally clear.